EMBO
Molecular Medicine

# Neonatal AAV gene therapy rescues hearing in a mouse model of *SYNE4* deafness

Shahar Taiber[1] (ID), Roie Cohen[2] (ID), Ofer Yizhar-Barnea[1] (ID), David Sprinzak[2] (ID), Jeffrey R Holt[3,*] (ID) &
Karen B Avraham[1,**] (ID)

## Abstract

**Genetic variants account for approximately half the cases of congenital and early-onset deafness. Methods and technologies for viral delivery of genes into the inner ear have evolved over the past decade to render gene therapy a viable and attractive approach for treatment. Variants in *SYNE4*, encoding the protein nesprin-4, a member of the linker of nucleoskeleton and cytoskeleton (LINC), lead to DFNB76 human deafness. *Syne4*$^{-/-}$ mice have severe-to-profound progressive hearing loss and exhibit mislocalization of hair cell nuclei and hair cell degeneration. We used AAV9-PHP.B, a recently developed synthetic adeno-associated virus, to deliver the coding sequence of *Syne4* into the inner ears of neonatal *Syne4*$^{-/-}$ mice. Here we report rescue of hair cell morphology and survival, nearly complete recovery of auditory function, and restoration of auditory-associated behaviors, without observed adverse effects. Uncertainties remain regarding the durability of the treatment and the time window for intervention in humans, but our results suggest that gene therapy has the potential to prevent hearing loss in humans with *SYNE4* mutations.**

**Keywords** deafness; DFNB76; gene therapy; Nesprin-4; SYNE4

**Subject Categories** Genetics, Gene Therapy & Genetic Disease; Neuroscience

## Introduction

Hearing loss affects approximately 466 million people worldwide (Olusanya *et al*, 2019). A genetic cause can be identified in 60% of the cases of hearing loss in multiplex families, and more than 120 genes have been associated with non-syndromic hearing loss in humans (Brownstein *et al*, 2020; Van Camp & Smith, 2020). Although tremendous progress has been made in the understanding of the physiology the auditory system, there are still no biological treatments for hearing loss in humans. Major efforts are currently

being made to develop gene, cell, and pharmacological therapeutics for various types of hearing loss, but current treatment options are still primarily restricted to sound amplification and cochlear implants (Muller & Barr-Gillespie, 2015; Schilder *et al*, 2018).

Variants in *SYNE4* (Spectrin Repeat Containing Nuclear Envelope Family Member 4) have been found to cause autosomal recessive progressive, high-tone hearing loss in individuals in Israel, the UK, and Turkey (PanelApp.; Horn *et al*, 2013; Masterson *et al*, 2018). *SYNE4* codes for the protein nesprin-4, a member of the linker of nucleoskeleton and cytoskeleton (LINC) complex (Roux *et al*, 2009). Nesprins localize to the outer nuclear membrane, where they interact with inner nuclear membrane SUN proteins, and with cytoplasmatic cytoskeleton elements such as actin and intermediate filaments, as well as motor proteins such as kinesins and dynein (Cartwright & Karakesisoglou, 2014). Mice lacking *Syne4* or *Sun1* exhibit progressive hearing loss, reminiscent of DFNB76; in *Syne4* knockout mice (*Syne4*$^{-/-}$), hair cells develop normally, but the outer hair cell (OHC) nuclei gradually lose their basal position, leading to subsequent OHC degeneration (Horn *et al*, 2013).

Preliminary results in animal models identified adeno-associated virus (AAV) as a promising candidate for gene therapy in deafness (Landegger *et al*, 2017; Akil *et al*, 2019; Isgrig *et al*, 2019; Nist-Lund *et al*, 2019). AAVs appear to elicit little to no immune response, and recombinant AAVs integrate into the host at very low rates, which reduces the risks of genotoxicity (Nakai *et al*, 2001). Initial characterization of natural AAV serotypes revealed a relatively low transduction rate of inner ear cell types, and in particular of OHC (Kilpatrick *et al*, 2011). However, recently developed synthetic AAV capsids seem to have overcome this hurdle; AAV9-PHP.B has been shown to transduce both inner and outer hair cells at high rates in mice and non-human primates (Gyorgy *et al*, 2019; Ivanchenko *et al*, 2020; Lee *et al*, 2020).

In this study, we used *Syne4*$^{-/-}$ mice as a model of DFNB76 recessive deafness, in order to develop a genetic therapy for this form of human deafness, based on AAV9-PHP.B as a vector. In addition to morphological recovery of transduced OHC, we observed enhanced OHC survival, improved auditory brainstem responses (ABR), and restored distortion-product otoacoustic emissions (DPOAE). In addition, we demonstrate that functional recovery of

1 Department of Human Molecular Genetics & Biochemistry, Sackler Faculty of Medicine & Sagol School of Neuroscience, Tel Aviv University, Tel Aviv, Israel
2 School of Neurobiology, Biochemistry and Biophysics, George S. Wise Faculty of Life Sciences, Tel Aviv University, Tel Aviv, Israel
3 Departments of Otolaryngology & Neurology, Boston Children's Hospital & Harvard Medical School, Boston, MA, USA
*Corresponding author. Tel: +1 617 919 3574; E-mail: jeffrey.holt@childrens.harvard.edu
**Corresponding author. Tel: +972 3 640 6642; Fax: +972 3 640 9360, E-mail: karena@tauex.tau.ac.il

the inner ear is sufficient to drive complex behavioral responses that rely on processing of auditory cues in the central nervous system. Finally, we characterize the safety of exogenous *Syne4* overexpression in both the auditory and vestibular systems. While the feasibility of translating these results to the clinic is still unclear, we conclude that our results in *Syne4*$^{-/-}$ mice suggest that gene therapy for DFNB76 is a future possibility that should be developed.

# Results

### *Syne4*$^{-/-}$ outer hair cells degenerate at hearing onset

Nesprin-4, which is encoded by the *Syne4* gene, has been shown to be important for nuclear positioning and OHC survival in mice (Horn *et al*, 2013). With the aim of developing gene therapy for *Syne4*$^{-/-}$ mice, we studied the dynamics of OHC loss in order to determine a therapeutic time window for intervention. A schematic illustration of the ear (Fig 1A), with a focus on the organ of Corti, as well as the timeline of the experiments performed in the study (Fig 1B), are shown. We analyzed hair cell survival in the inner ear at P8, P10, P12, and P14 (Figs 2A and B, and EV1). While at P8, the OHCs appeared intact, by P14, their degeneration was readily apparent (Fig 2A and B). This was also reflected in hair cell counts, with an apex-to-base gradient in the reduction of the number of OHCs by P14 (Fig 2C). FM1-43 is a styryl dye that can enter hair cells through the sensory transduction channels (Gale *et al*, 2001) and is used as a proxy for functional hair cell sensory transduction. We found that *Syne4*$^{-/-}$ hair cells could take up FM1-43 at P8, in a similar manner to wild-type (WT) hair cells (Fig 2D). This suggests that *Syne4*$^{-/-}$ hair cells mature and acquire the properties of functional hair cells prior to the onset of damage. In addition, we examined the expression pattern of *Syne4* in published datasets via the gEAR portal (Portal) and found *Syne4* expression to be relatively

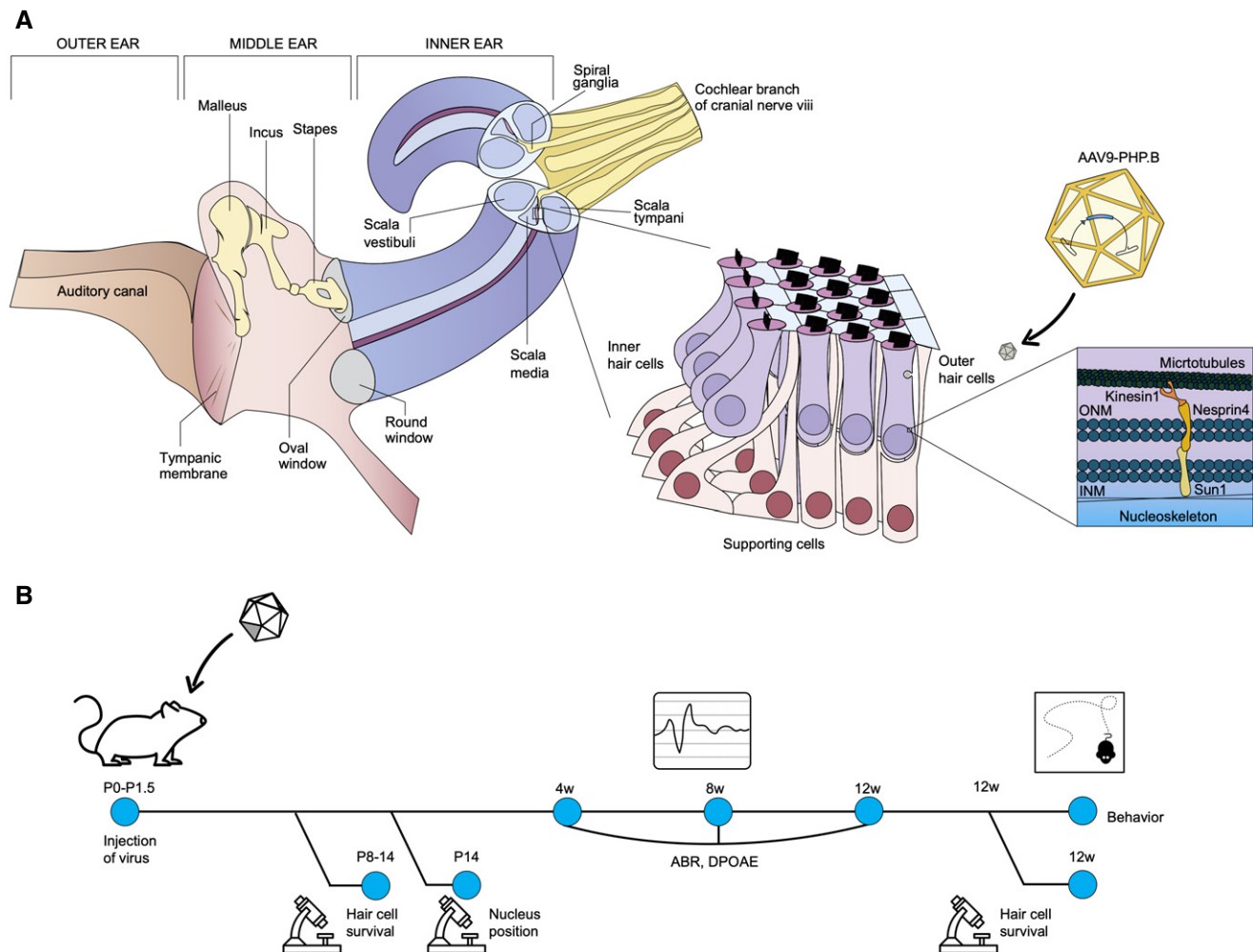

**Figure 1. Schematic representation of research strategy and timeline.**

A   Schematic representation of inner ear anatomy, with a focus on the organ of Corti and the cellular function of nesprin-4.

B   Timeline of experiments performed.

restricted to hair cells, with a higher expression in OHCs than inner hair cells (IHC) (Fig EV2A; Scheffer *et al*, 2015; Liu *et al*, 2018). *Syne4* RNA is detected as early as P0, although the staining at P0 is weaker than at P12 (Horn *et al*, 2013, Fig EV2B). Interestingly, while *Syne4* is also detected in the vestibular system, *Syne4*$^{-/-}$ mice exhibited no abnormal balance behavior (Fig EV2B and C).

### AAV9-PHP.B transduces cochlear hair cells in neonatal mice

AAV9-PHP.B is a synthetic AAV capsid that has been engineered by directed in-vivo evolution (Deverman *et al*, 2016) and transduces both IHCs and OHCs at high rates (Gyorgy *et al*, 2019; Lee *et al*, 2020). Expression of GFP delivered in AAV9-PHP.B begins rising reliably between days 3 and 5 postinjection (Lee *et al*, 2020). We therefore chose this capsid as a vector for gene therapy in *Syne4*$^{-/-}$ mice. We cloned turboGFP into an AAV2 backbone, downstream of a CMV enhancer and promoter and upstream to a bGH poly-A sequence, and packaged the construct into AAV9-PHP.B capsids (termed AAV.GFP). We then cloned the coding sequence (CDS) of *Syne4* into an AAV2 backbone, downstream of a CMV enhancer and promoter, added a 3XFLAG epitope sequence at the 5′ end of the

*Syne4* CDS and a bGH poly-A sequence at the 3′ and packaged the construct into AAV9-PHP.B capsids (termed AAV.Syne4) (Fig 3A). The titers of AAV.Syne4 and AAV.GFP were 7.7E + 12 gc/ml and 8.6E + 12 gc/ml, respectively.

In order to examine the transduction efficiency of this capsid, we injected WT mice at P0–P1.5 with AAV.GFP, using the previously described posterior-semicircular canal (PSCC) approach for inner ear delivery (Isgrig & Chien, 2018). Inner ears were harvested at P9 for immunofluorescence and quantification of transduction rate (Fig 3B). Comparing the results to an un-injected littermate as control for background fluorescence, we observed GFP expression in all IHCs and OHCs in the 8, 16, and 32 kHz regions of the organ of Corti, as well as strong fluorescence in Deiters cells, pillar cells, and Hensen's cells (Fig 3C). The normalized GFP intensity was similar between three individual injected mice and typically higher in OHC as compared to IHC (Fig 3D). Cells were regarded as GFP positive if the GFP intensity was higher than 2 standard deviations above the average intensity measured in the control mouse. Despite the transduction rate being 100% in all three regions of the cochlea in all three mice examined (Fig 3E), IHC fluorescence was lower than OHC (Fig 3D).

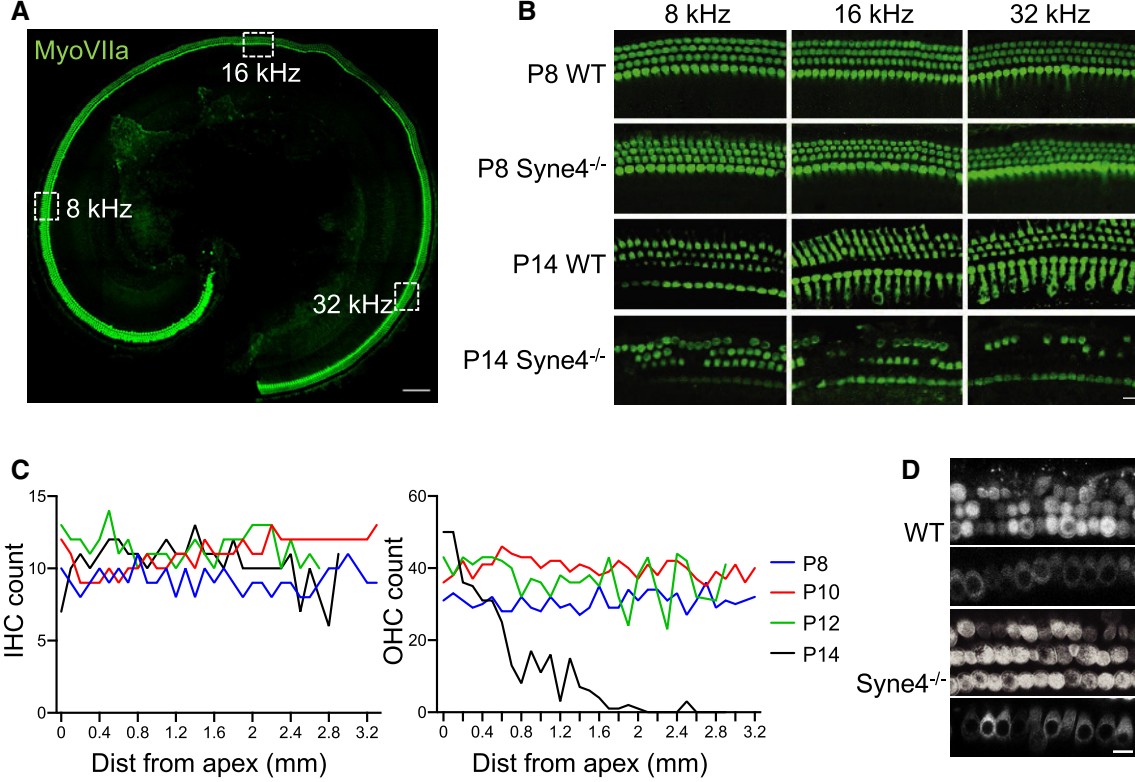

**Figure 2. *Syne4*$^{-/-}$ hair cells mature normally and then degenerate between P12 and P14.**

A Whole-mount immunofluorescence of a P8 *Syne4*$^{-/-}$ organ of Corti showing intact hair cells, as labeled with myosin VIIa.

B Whole-mount immunofluorescence of WT and *Syne4*$^{-/-}$ organ of Corti from the 8, 16, and 32 kHz regions at P8 and P14.

C Inner and outer hair cell counts of *Syne4*$^{-/-}$ organ of Corti at P8, P10, P12, and P14.

D FM1-43 uptake performed on P8+1 DIV (days-*in-vitro*) WT and *Syne4*$^{-/-}$ organ of Corti. Top shows OHC plane, and bottom shows IHC plane.

Data information: Scale bars = 100 μm for (A) and 10 μm for (B and D).
Source data are available online for this figure.

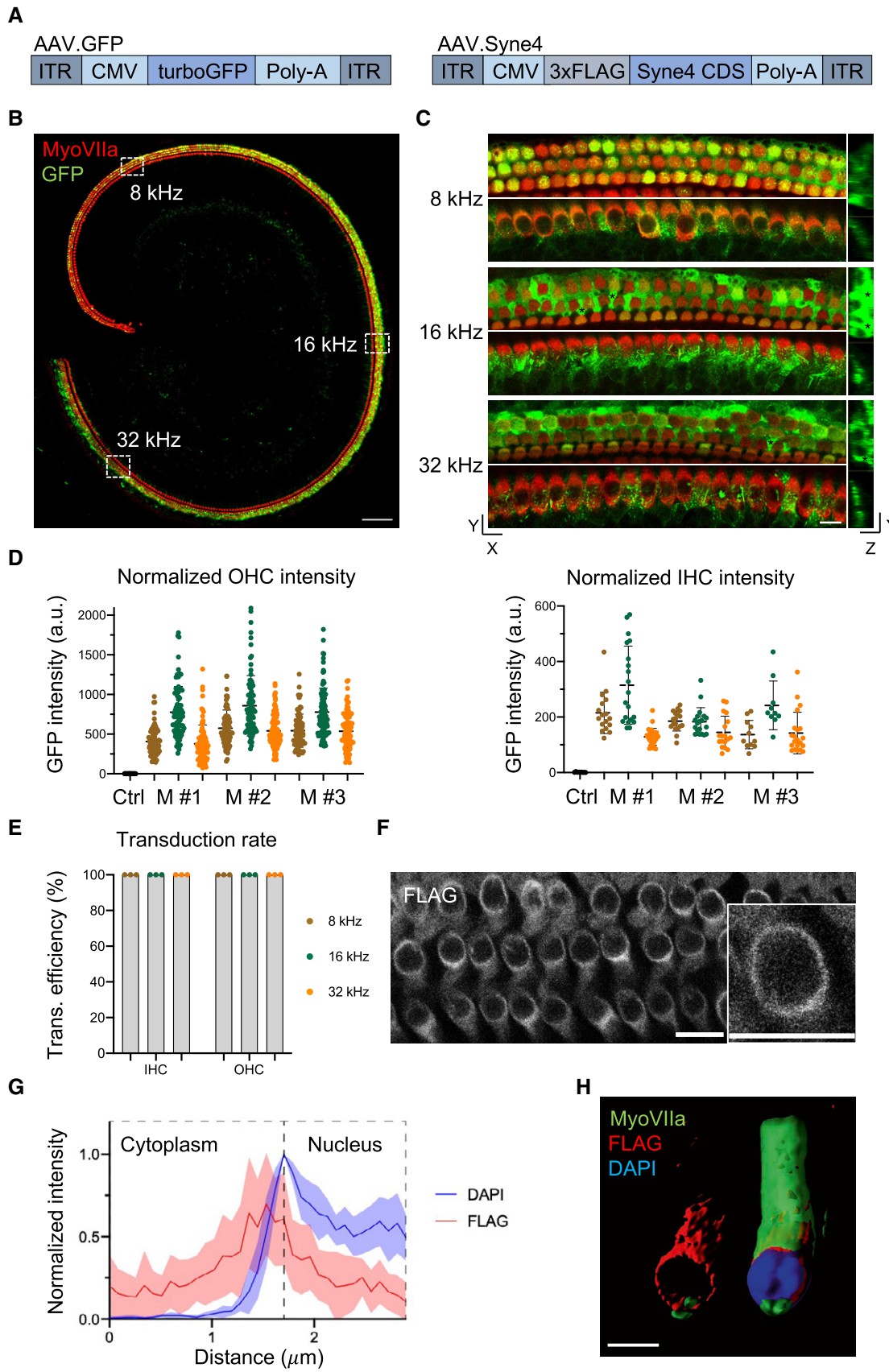

Figure 3.

**Figure 3.  AAV9-PHP.B transduction and *Syne4* expression.**

A   Schematic representation of AAV.Syne4 and AAV.GFP constructs.

B   Whole-mount immunofluorescence of a P9 organ of Corti of a mouse injected with AAV.GFP at P1 showing complete transduction of both inner and outer HC. Myosin VIIa was used to label the hair cells.

C   Examples of 8, 16, and 32 kHz regions of an organ of Corti of a mouse injected with AAV.GFP. Top shows OHC plane, bottom shows IHC plane, and right shows YZ orthogonal projection. Black asterisks show bright Deiters cells.

D   Quantification of GFP intensity of inner and outer HC from 3 injected mice, normalized to the average intensity of HC in a control, un-injected mouse.

E   Transduction rates of AAV9-PHP.B at 8, 16, and 32 kHz regions based on GFP fluorescence. A total of 162 IHC and 841 OHC were analyzed from 3 injected mice and 1 control littermate.

F   Staining for FLAG at P14 of the organ of Corti of a mouse injected at P1 with AAV.Syne4.

G   Quantification of FLAG and DAPI fluorescence intensity along a line centered at the nuclear envelope. Eight OHCs were measured.

H   3D surface projection of two adjacent OHC from a mouse injected with AAV.Syne4. In the left cell, myosin VIIa and DAPI signals were removed to only show FLAG staining.

Data information: Scale bars = 100 μm for (B), 10 μm for (C and F), and 5 μm for (H). Plots show mean ± SD.
Source data are available online for this figure.

In addition, we injected mice with AAV.Syne4 and stained the ears for FLAG. We observed FLAG staining in a pattern that indicated nesprin-4 was localized to the nuclear envelope (Fig 3F–H), as described previously for endogenous nesprin-4 (Roux *et al*, 2009; Horn *et al*, 2013), suggesting that the 3XFLAG-nesprin-4 protein was folded correctly.

**Viral transduction and *Syne4* overexpression not associated with long-term ototoxicity or vestibulotoxicity**

To test the safety of the AAV9-PHP.B capsid, as well as the overexpression of exogenous *Syne4*, we injected WT mice with AAV.Syne4 at P0-P1.5, and evaluated ABR and DPOAE at 4, 8, and 12 weeks. Injected mice showed no significant difference from control mice in ABR threshold values ($P > 0.58$ for all frequencies tested at 4 weeks, $P > 0.25$ for all frequencies tested at 8 weeks, and $P > 0.49$ for all frequencies tested at 12 weeks, Fig EV3A–C). DPOAE thresholds at 4w were also not significantly different ($P > 0.89$ for all frequencies tested) (Fig EV3D). Open-field tests were performed at 12 weeks to exclude possible toxicity to the vestibular system. Injected mice showed no overt balance defects ($P > 0.19$ for the three parameters tested) (Fig EV3E). Finally, weight gain, which was used as a measurement of general health, remained unchanged in injected mice ($P > 0.7$ for WT mice compared to WT mice injected with AAV.Syne4 at all time points, Fig EV3F).

**AAV.Syne4 prevents nuclear mislocalization in *Syne4*$^{-/-}$ outer hair cells**

To test the effect of virally mediated expression of *Syne4* on hair cell morphology, we injected *Syne4*$^{-/-}$ mice with AAV.Syne4 at P0–P1.5 and harvested inner ears at P14 (Fig 4). In *Syne4*$^{-/-}$ mice, the nuclei of OHCs are mislocalized and are positioned close to the cuticular plate (Fig 4A and B). Hair cell nuclear position was quantified semi-automatically, according to the length of an arc measured between the apical and basal ends of the cell and the position of the nucleus along that arc (Fig 4C). As compared to un-injected *Syne4*$^{-/-}$ mice, the OHC nuclei in injected mice were situated closer to the base, but were not identical to the situation in WT mice (Fig 4D). In contrast, the nuclear position in the IHCs was not affected by either the mutation or the treatment (Fig 4D).

**AAV.Syne4 rescues auditory function to near wild-type levels**

To evaluate the therapeutic effect of *Syne4* delivery, we injected *Syne4*$^{-/-}$ mice at P0-P1.5 with AAV.Syne4 or AAV.GFP as a control, and assessed the ABR and DPOAE at 4, 8, and 12 weeks (Figs 5 and EV4). For two mice, there was no evidence of success of the injection, as evaluated by immunofluorescence analysis of *Syne4* expression or ABR/DPOAE recovery, and thus, they were excluded from downstream analyses. The results revealed that mice injected with AAV.Syne4 ($n = 20$) had fully restored auditory function at 4 weeks that was sometimes indistinguishable from WT controls ($P < 0.0001$ for injected *Syne4*$^{-/-}$ mice, as compared to *Syne4*$^{-/-}$ mice injected with AAV.GFP or un-injected) (Fig 5A and B). At 4 weeks, the amplitudes and latencies of the response to a 0.1 ms click stimulus as a function of stimulus intensity were also highly similar to those of WT controls (Fig 5C and D). DPOAE assessments at 4 weeks showed complete recovery of thresholds ($P < 0.0001$ for injected *Syne4*$^{-/-}$ mice compared to un-injected *Syne4*$^{-/-}$ mice at 12.4–3.5 kHz, $P = 0.22$ for 6 kHz, $P > 0.53$ for injected *Syne4*$^{-/-}$ mice compared to wild-type mice for all frequencies tested). This finding suggests that the OHCs of treated mice were functional (Fig 5E).

At 12 weeks, the ABR thresholds of treated mice were already significantly higher than those of WT mice, but still significantly lower than *Syne4*$^{-/-}$ mice injected with AAV.GFP or un-injected ($P < 0.05$ for injected *Syne4*$^{-/-}$ mice compared to WT mice for 12–35 kHz, $P = 0.055$ for 6 kHz, $P < 0.0001$ for 6–24 kHz, $P = 0.001$ for 30 kHz, and $P = 0.032$ for 35 kHz for *Syne4*$^{-/-}$ mice compared to *Syne4*$^{-/-}$ mice injected with AAV.Syne4) (Fig 5F). It is worth noting that while the parameters in some mice deteriorated over time, others did not, and that the best performing mouse in the injected group still had low ABR thresholds at 12 weeks (30, 15, 15, 30, 35, and 45 dB-SPL for 6, 12, 18, 24, 30, and 35 kHz, respectively).

**AAV.Syne4 promotes long-term outer hair cell survival**

In order to examine the effect of treatment on hair cell survival, we counted myosin VIIa-positive cells along the length of the cochlea. The results indicated survival of virtually all OHC in treated *Syne4*$^{-/-}$ mice, lasting up to 12 weeks postinjection, but no apparent difference in IHC survival between the three groups (Fig 6A–C). To further validate our hypothesis that *Syne4* deafness stems predominantly from OHC loss and not impaired IHC function, we

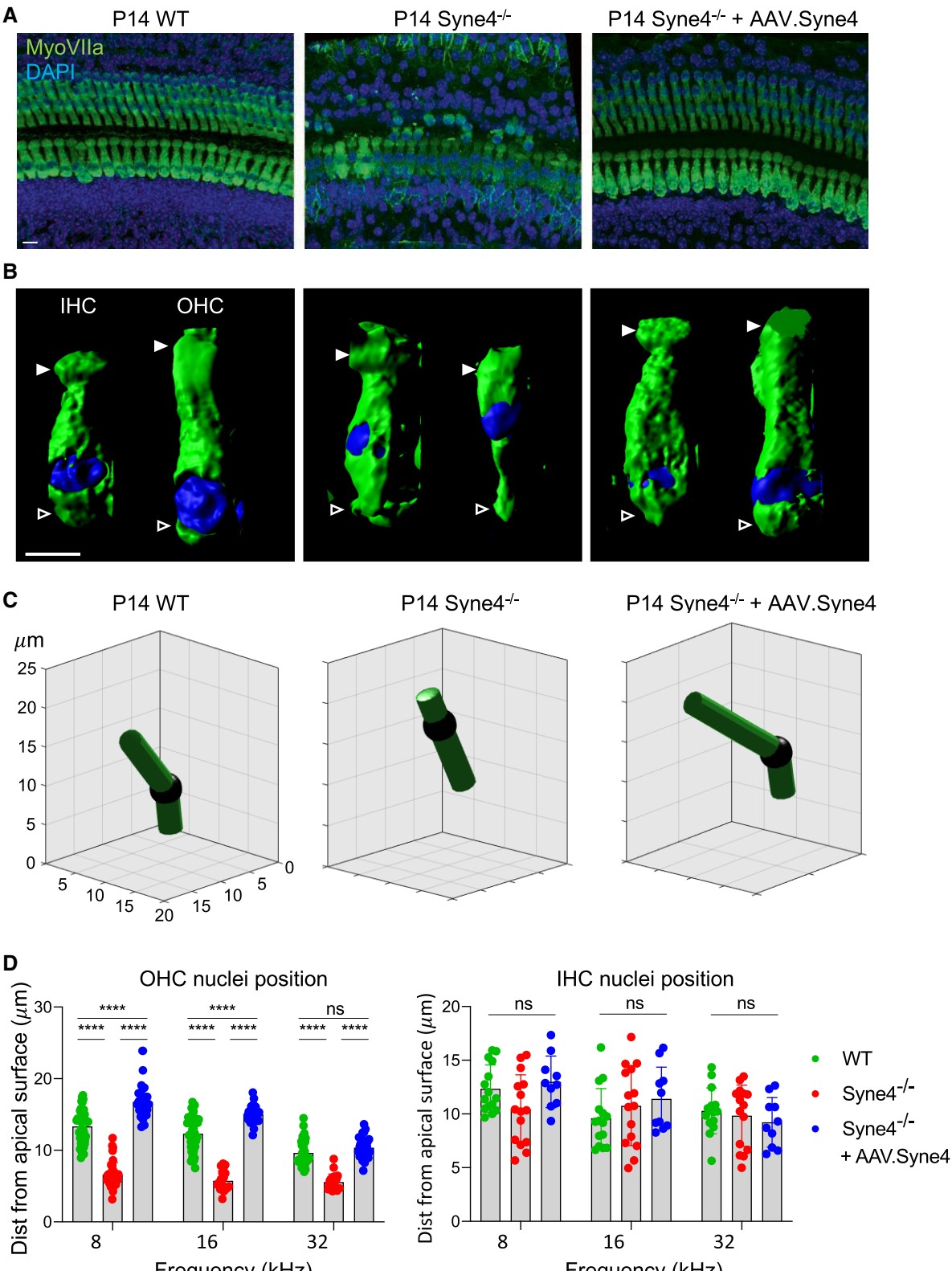

**Figure 4.**

**Figure 4.** AAV.Syne4 rescues OHC morphology in *Syne4*[−/−] mice.

A  Whole-mount immunofluorescence of the 8 kHz region from P14 organ of Corti from WT, *Syne4*[−/−], and *Syne4*[−/−] mice injected with AAV.Syne4.

B  3D surface projection of IHC and OHC at P14 from the 8kHz region of WT, *Syne4*[−/−] mice and *Syne4*[−/−] mice injected with AAV.Syne4. Open arrows denote basal end, and white arrows denote cuticular plate.

C  Image analysis of nucleus position of IHC and OHC. Images show an arc fitted through the apical surface, nucleus, and basal end of the cell in 3D. X, Y, and Z axes show position in μm.

D  Quantification of nuclear distance from the cuticular plate. A total of 45 IHC and 135 OHC from 3 WT mice, 45 IHC and 81 OHC from 3 *Syne4*[−/−] mice, and 30 IHC and 90 OHC from 2 *Syne4*[−/−] mice injected with AAV.Syne4 were measured. Statistical test was 2-way ANOVA with Holm–Sidak correction for multiple comparisons. Plots show mean ± SD. ns = not significant, ****$P < 0.0001$.

Data information: Scale bars = 10 μm for (A) and 5 μm for (B).
Source data are available online for this figure.

examined whether the variability in ABR threshold levels at 12 weeks could be explained by OHC survival. We found a strong negative correlation between the numbers of OHCs per 100 μm and the ABR threshold in the 12 kHz region of the organ of Corti ($r = -0.89$, $P = 0.0003$), which we defined as 1.63–1.69 mm from the apex, based on a place-frequency map of the cochlea (Müller *et al*, 2005) (Fig 6D). Since there was little change in IHC survival, the number of IHC was not correlated with the ABR threshold ($r = -0.29$, $P = 0.36$) (Fig 6D). We tested whether the observed deterioration of auditory function in some mice in the treatment group could be explained by a change in the position of the nuclei that does not lead to OHC death but does impair their function. For this purpose, we quantified the position of OHC nuclei at the 12 kHz region of the organ of Corti at 12 weeks (Fig 6E–G). We could not detect a significant change in their position at 12 weeks, suggesting that this could not explain the deterioration we observed in some of the treated mice.

**AAV.Syne4 rescues behavioral responses to auditory stimuli**

A number of deafness genes have been shown to function in the central auditory system (Kharkovets *et al*, 2000; Libe-Philippot *et al*, 2017) and at the RNA level, *Syne4* is expressed in parts of the central nervous system (GTEx Portal). Therefore, *Syne4* could have a role in the function of the central auditory system and peripheral delivery of *Syne4* might not rescue auditory functions that rely on central processing of sound. In addition, damage to the peripheral auditory system has long been known to impair the development of the central auditory system as a secondary consequence (Levi-Montalcini, 1949; Gilley *et al*, 2008; Sharma *et al*, 2009). To investigate whether peripheral delivery of exogenous *Syne4* into *Syne4*[−/−] mice would be sufficient to also rescue auditory behaviors that require central auditory processing, we used cued fear conditioning (Fig 7A, Movie EV1). *Syne4*[−/−] mice injected with AAV.Syne4 or AAV.GFP, as well as untreated, and WT mice, were positioned in a cage inside an acoustic chamber. A 6 kHz tone pip was followed by a short electric shock delivered through the cage. This step was repeated twice on day 1. The next day, mice were again placed in the same cage and presented with the same tone, although this time no shock was delivered. A camera was used to quantify the movement of mice, identify the activity level, and detect freezing behavior, which is indicative of fear and memory that rely on central processing of the auditory stimulus (Ciocchi *et al*, 2010; Weinberger, 2011; Courtin *et al*, 2013; Courtin *et al*, 2014). Both *Syne4*[−/−] mice injected with AAV.GFP and untreated *Syne4*[−/−] mice seem to freeze at similar rates regardless of the appearance of the stimulus

($P > 0.16$) (Fig 7B). Since the mice could not hear the stimulus, they probably associated the shock with other clues found in the scene, such as the smells and appearances of the cage, the room, and the tester. In contrast, both *Syne4*[−/−] mice injected with AAV.Syne4, and WT mice exhibited significantly more freezing behavior during the tone period ($P < 0.0001$) (Fig 7B). In addition, activity levels during the tone period were significantly lower in WT mice and *Syne4*[−/−] mice injected with AAV.Syne4 ($P \leq 0.0005$), while no decrease in activity level was detected in *Syne4*[−/−] mice injected with AAV.GFP and untreated *Syne4*[−/−] mice ($P = 0.8609$) (Fig 7C). This indicates that the mice could hear and process the stimulus and were able to associate the shock with the tone.

## Discussion

We have previously reported that both humans with loss-of-function variants in *SYNE4* (DFNB76) and *Syne4*[−/−] mice have progressive hearing loss. We also demonstrated that Sun1, a nesprin-interacting member of the LINC complex, is necessary for hearing in mice (Horn *et al*, 2013). More cases of *SYNE4* deafness in humans have since been reported in Turkey (Masterson *et al*, 2018) and the UK (PanelApp.). Taken together, these observations imply that LINC complex proteins play a role in the function of hair cells and suggest that more cases of LINC complex-associated deafness may exist. We therefore sought to test whether a gene therapy approach using *Syne4*[−/−] mice as a model and AAV9-PHP.B as a delivery vector could be used to rescue hearing in such cases. In agreement with previous reports on the use of AAV9-PHP.B in the inner ear, we found robust expression, with transduction of all inner and outer hair cells along the cochlea. However, GFP levels in IHC were low in comparison to OHC. It is possible that the rescue we observed should only be attributed to OHC transduction, which would strengthen our hypothesis that *Syne4* deafness stems primarily from OHC dysfunction and degeneration. Differences between our observed transduction rates and previous reports may be explained by variations in injected titer, delivery route, the use of different GFP variants, and the quantification method. We utilized PSCC injection as a method of delivery and injected mice at P0–P1.5, before the onset of hearing, and before pathological morphological changes occur in the hair cells. It is not clear whether a later intervention would still be relevant, but since the majority of OHC in these mice rapidly degenerate at the onset of hearing, and since several days pass between injection and expression of the transgene (Lee *et al*, 2020), we believe that the time window may be restricted. While the onset of hearing in humans is toward the end

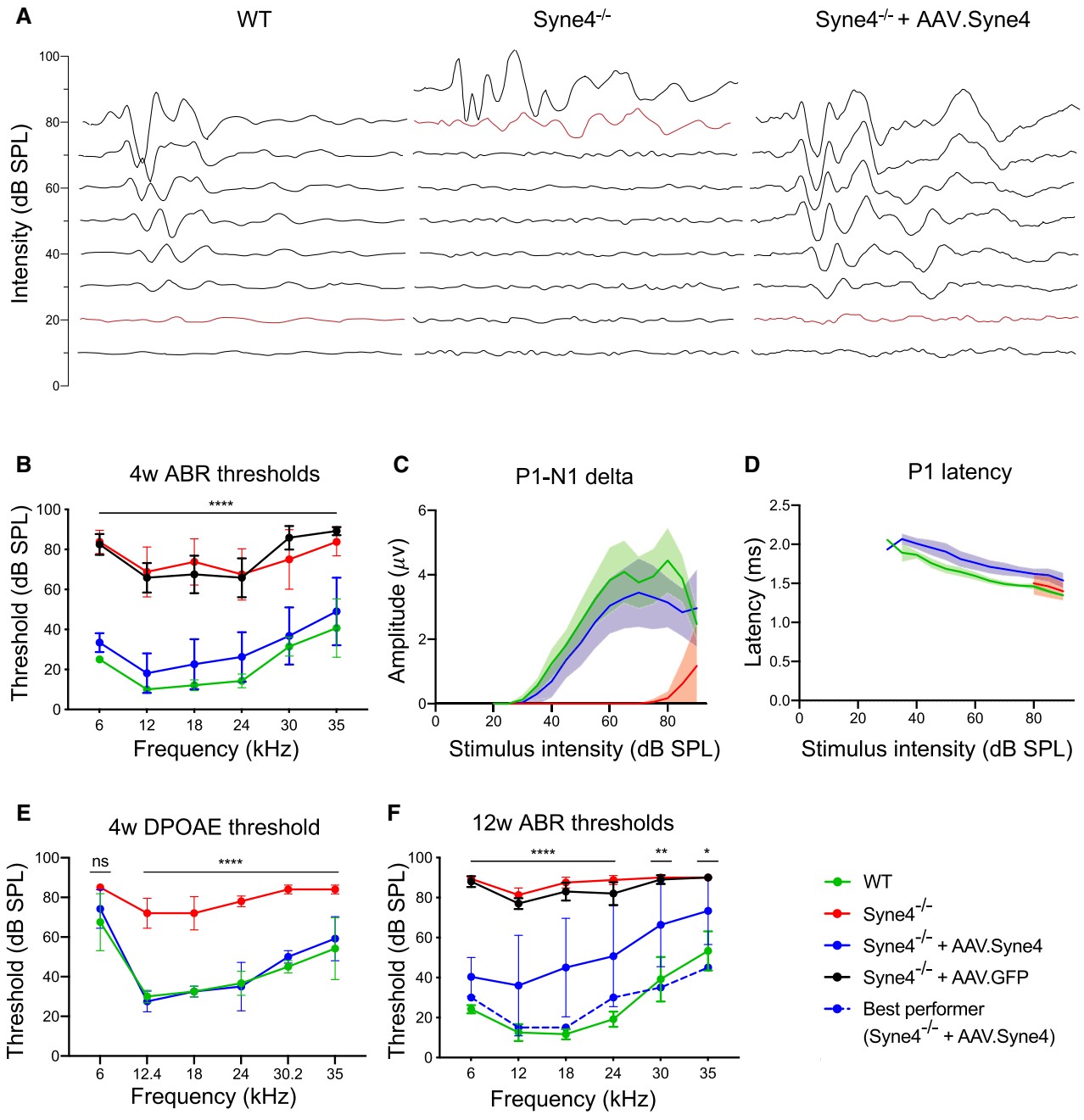

**Figure 5. AAV.Syne4 rescues auditory function in *Syne4*$^{-/-}$ mice.**

A Representative example of ABR traces obtained at 4w from a WT, *Syne4*$^{-/-}$ mouse, and *Syne4*$^{-/-}$ mouse injected with AAV.Syne4 in response to 18 kHz stimuli.

B ABR thresholds at 4w of WT, *Syne4*$^{-/-}$, *Syne4*$^{-/-}$ mice injected with AAV.Syne4, and *Syne4*$^{-/-}$ mice injected with AAV.GFP, $n = 7$ for WT, $n = 8$ for *Syne4*$^{-/-}$, $n = 20$ for *Syne4*$^{-/-}$ + AAV.Syne4, and $n = 6$ for *Syne4*$^{-/-}$ + AAV.GFP.

C Quantification of P1-N1 amplitude delta from (B), $n = 4$ for WT, $n = 4$ for *Syne4*$^{-/-}$, and $n = 7$ for *Syne4*$^{-/-}$ + AAV.Syne4.

D Quantification of P1 latency from (B), $n = 4$ for WT, $n = 2$ for *Syne4*$^{-/-}$, and $n = 7$ for *Syne4*$^{-/-}$ + AAV.Syne4.

E DPOAE thresholds at 4w, $n = 6$ for WT, $n = 5$ for *Syne4*$^{-/-}$, and $n = 6$ for *Syne4*$^{-/-}$ + AAV.Syne4.

F ABR threshold at 12w, $n = 6$ for WT, $n = 8$ for *Syne4*$^{-/-}$, $n = 16$ for *Syne4*$^{-/-}$ + AAV.Syne4, and $n = 5$ for *Syne4*$^{-/-}$ + AAV.GFP.

Data information: Statistical test was 2-way ANOVA with Holm–Sidak correction for multiple comparisons. Plots show mean ± SD. ns = not significant, *$P < 0.05$, **$P < 0.01$ ****$P < 0.0001$.

Source data are available online for this figure.

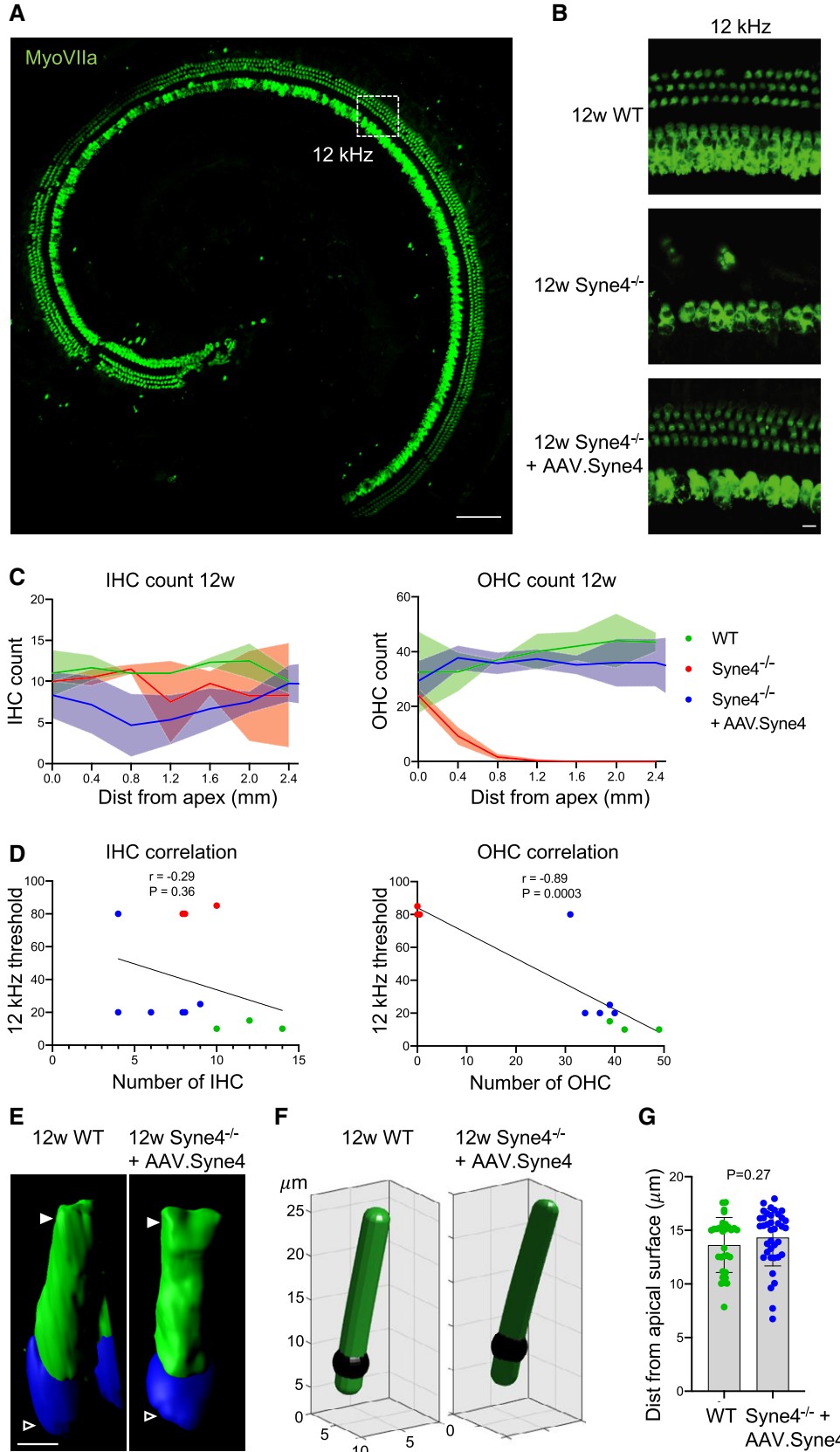

**Figure 6.**

**Figure 6.   AAV.Syne4 promotes long-term survival of OHC in *Syne4*<sup>−/−</sup> mice.**

A     Whole-mount immunofluorescence at 12w of the organ of Corti of a *Syne4*<sup>−/−</sup> mouse injected with AAV.Syne4 mouse injected with AAV.Syne4.
B     Whole-mount immunofluorescence at 12w of the 12 kHz region of WT, *Syne4*<sup>−/−</sup> mouse, and *Syne4*<sup>−/−</sup> mouse injected with AAV.Syne4.
C     IHC and OHC counts, *n* = 3 for WT, *n* = 4 for Mut, and *n* = 6 for Mut + AAV.Syne4.
D     Correlation between HC count and ABR threshold at 12kHz.
E     3D surface projection of an OHC at 12w from the 12kHz region of WT and *Syne4*<sup>−/−</sup> mice injected with AAV.Syne4. Open arrows denote basal end, and white arrows
        denote cuticular plate.
F     Image analysis of nucleus position of OHC.
G     Quantification of nuclear distance from the cuticular plate. A total of 30 OHC from 3 WT mice and 38 OHC from 4 *Syne4*<sup>−/−</sup> mice injected with AAV.Syne4 were
        measured.

Data information: Scale bars = 100 μm for (A), 10 μm for (B), and 5 μm for (E). Statistical tests were Pearson correlation with two-tailed *P* values for (D) and unpaired
Student's *t*-test for (G). Plots show mean ± SD.
Source data are available online for this figure.

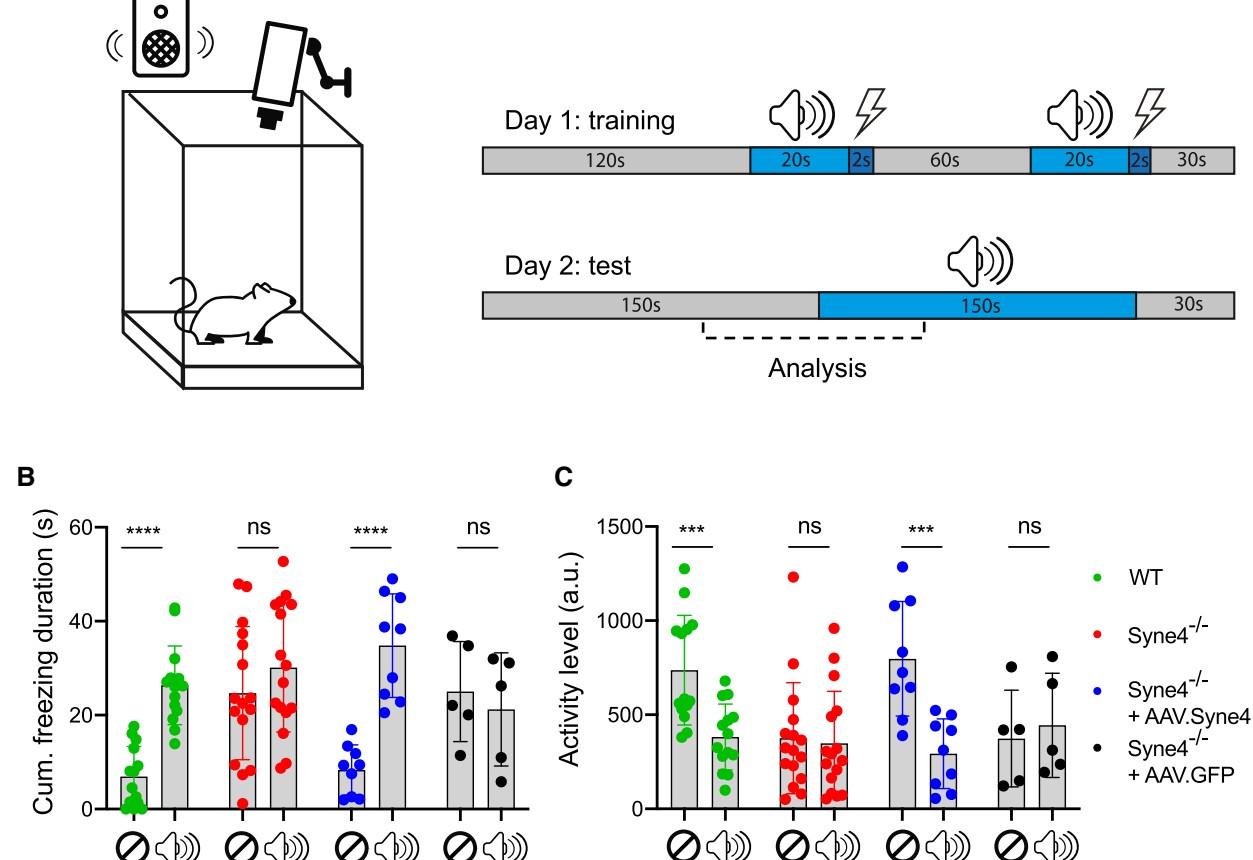

**Figure 7.   AAV.Syne4 rescues behavioral response to sound in *Syne4*<sup>−/−</sup> mice.**

A     Schematic representation of cued fear conditioning experimental procedure.
B, C     Quantification of activity level and cumulative freezing duration at 12w during seconds 90–210 of day 2 in WT, *Syne4*<sup>−/−</sup> mice, *Syne4*<sup>−/−</sup> mice injected with
        AAV.Syne4, and *Syne4*<sup>−/−</sup> mice injected with AAV.GFP, *n* = 14 for WT, *n* = 16 for *Syne4*<sup>−/−</sup>, *n* = 9 for *Syne4*<sup>−/−</sup> + AAV.Syne4, and *n* = 5 for *Syne4*<sup>−/−</sup> + AAV.GFP. (C)
        Activity level. (B) Cumulative freezing duration. Statistical test was repeated-measures ANOVA with Holm–Sidak correction for multiple comparisons, comparing the
        scores of each individual mouse before and after the appearance of the tone. Plots show mean ± SD. ***$P < 0.001$, ****$P < 0.0001$. ns = not significant.

Source data are available online for this figure.

of the first trimester of pregnancy (Gagnon *et al*, 1986), humans with variants in *SYNE4* exhibit a postnatal onset of hearing loss, with a much more gradual progression than seen in *Syne4*<sup>−/−</sup> mice. We therefore believe that this observation corresponds to a wide therapeutic time window in humans.

With regard to potential toxicity caused by high gene dosage or by expression of the transgene in other cell populations, such as cochlear supporting cells and vestibular hair cells, we observed no general adverse effect on hearing or balance function of WT mice injected with the virus. Neither was there any change in weight

gain. However, in future studies it will be important to characterize the systemic response to the virus more comprehensively, for example, by evaluating the production of neutralizing antibodies against the viral capsid proteins.

Our results indicate that exogenous delivery of *Syne4* into neonatal *Syne4*$^{-/-}$ hair cells rescues their morphology and improves survival. Nuclei of OHCs from injected ears were positioned at the basal part of the cell, as in WT mice, and there was no observed difference in long-term OHC survival or nuclei position. We found that *Syne4* delivery also resulted in near-complete rescue of ABR and DPOAE thresholds, with best performers showing ABR thresholds as low as 15 dB for some frequencies. These thresholds are considered normal in humans. However, in some mice, ABR thresholds deteriorated over time. We are not sure whether this is due to silencing of the transgene, the result of an immune reaction against the viral capsid or the transgene or must be attributed to another cause. We did not observe substantial loss of OHC at 12 weeks; nor did we observe any significant change in nuclear position that could explain why some mice performed less well than others. It is possible that repeated dosing of AAV.Syne4 may be needed to maintain auditory function, or, alternatively that deterioration occurred as a consequence of impaired viral delivery or low transduction rate. In the latter cases, better standardization of the procedures should lead to generally better outcomes. Further characterization and optimization of these parameters will be required to translate these results to the clinic.

Stimulation of the peripheral sensory apparatuses has long been known to be required for normal development of central sensory modules (Levi-Montalcini, 1949). Although *Syne4* is expressed in the CNS, it is not clear whether *Syne4* is necessary for central auditory function, as is the case for some deafness genes (Kharkovets *et al*, 2000; Libe-Philippot *et al*, 2017). For this reason, it was interesting to examine whether the observed physiological recovery of the inner ear and auditory brainstem could drive complex behavioral responses that rely on comprehensive central processing. We chose cued fear conditioning as a relatively simple, straight-forward assay by which we could test whether the mice could perceive and then act on sound. This assay has been shown to reflect the activity of central auditory pathways in the CNS (Ciocchi *et al*, 2010; Weinberger, 2011). As expected, untreated *Syne4*$^{-/-}$ mice and *Syne4*$^{-/-}$ mice injected with AAV.GFP froze on day 2 after fear induction, but this behavior was not correlated with administration of the sound stimulus. This indicates that memory was formed but could not be associated with the stimulus. Somewhat surprisingly, we found that injected *Syne4*$^{-/-}$ mice and WT mice did not freeze until exposed to the stimulus, suggesting that fear was associated exclusively with the sound stimulus and that the association with the context had been "overridden". Our results do not support a central role for *Syne4*, but rather suggest that it is only necessary for normal organ of Corti function. If the human auditory system is similar to that of the mouse in this respect, we predict that local delivery of *SYNE4* into the inner ear should be sufficient for recovery of auditory function.

There are currently over 120 genes associated with non-syndromic hearing loss in humans. While some variants are prevalent in certain populations, such as *GJB2* 167delT in Ashkenazi Jews (Sobe *et al*, 1999) and TMC1 p.Ser647Phe in Moroccan Jews (Brownstein *et al*, 2011), many of the deafness genes affect only a

handful of families. This raises concerns regarding the justification and feasibility of developing personalized treatments for these forms of deafness. However, collectively, rare variants account for a substantial proportion of patients with deafness. To more broadly address the needs of hearing loss patients, therapeutic solutions will also be required for the less prevalent variants. While a translational gap exists for treatment of rare diseases (Tambuyzer *et al*, 2020), given the extent of recovery we observed in this study, we believe that gene therapy for *SYNE4* deafness is not only feasible, but also imperative.

## Materials and Methods

### Mice

Mice were housed in a controlled temperature environment on a 12-h light–dark cycle. Food and water were provided ad libitum. Both males and females were used. Mouse ages were P8, P9, P10, P12, P14, 4w, 8w, and 12w. For every experiment, the age is specified in the figure legend. *Syne4*$^{-/-}$ mice were maintained on a C57Bl/6J background. Genotyping was performed on DNA prepared from ear-punch biopsies, and extracted and amplified using the KAPA HotStart Mouse Genotyping Kit (Sigma, KK7352). Genotyping primers for the WT allele were WT_FWD (5′-ACTCCCAGCTCCAAGCTACA-3′) and WT_REV (5′-GCAGAGCCAAAGAAACCAAG-3′), and for the galactosidase gene were LacZ_FWD (5′-GTCTCGTTGCTGCATAAACC-3′) and LacZ_REV (5′-TCGTCTGCTCATCCATGACC-3′). Cycling conditions were an initial 3-min denaturation at 95°C followed by 35 cycles of 30 s 95°C, 30 s 60°C, and 30 s at 72°C, with a final elongation of 3 min at 72°C. PCR products were loaded into a 2% agarose ethidium-bromide gel for electrophoresis.

### AAV production

AAV viral vectors were prepared by the Vector Core at Boston Children's Hospital, as described previously (Lee *et al*, 2020). AAV2.CMV.3xFLAG.Syne4.bGH and AAV2.CMV.turboGFP.bGH vector plasmids were cloned and transferred together with AAV9-PHP.B plasmid to the Vector Core for production of AAV2/9.PHP.B.CMV.3xFLAG.Syne4.bGH and AAV2/9.PHPB.CMV.turboGFP.bGH. Viral titers were calculated based on qPCR amplification with primers directed at the AAV2 ITR sequences. The AAV2/9.PHP.B.CMV.3xFLAG.Syne4.bGH titer was 7.7E + 12 gc/ml, and the AAV2/9.PHP.B.CMV.turboGFP.bGH titer was 8.6E + 12 gc/ml. Vectors were aliquoted into 10 μl vials and stored at −80°C until use.

### Animal surgery

A posterior-semicircular canal (PSCC) injection was carried out in mice at P0–P1.5. Mice were anesthetized by induced hypothermia and kept on a cold surface throughout the procedure. Surgery was performed under an operating binocular. After disinfecting the skin with povidone iodine and ethanol, a 1–2 mm incision was made in the left postauricular region covering the temporal bone. Underlying soft tissue was carefully dissected to expose the PSCC. Virus preparation (1.0–1.2 μl) was aspirated into a borosilicate glass pipette

(Drummond, Broomall, PA 2-000-100) pulled with a P-30 vertical micropipette puller (Sutter Instrument, Novato, CA). Glass pipettes were held by a stereotaxic device and connected to a CMA 102 Microdialysis Pump (CMA, Sweden). Once identified, the PSCC was gently punctured and the virus was microinjected for ∼ 2 min (∼ 10 nl/s). After all the virus was injected, the pipette was left in place for an additional 30 s before removing it. The skin was closed with a single 8-0 polypropylene suture and 5% lidocaine cream was applied for pain control, together with Carprofen (2 mg/kg) once daily for three days. After surgery, mice were placed on a heating pad for recovery and were thoroughly cleaned of remaining iodine and ethanol before returning them to their mothers. Total surgery time did not exceed 15 min.

### Auditory testing

ABR and DPOAE, measurements were performed on mice anesthetized by intra-peritoneal injection of a combination of ketamine (100 mg/kg) and xylazine (10 mg/kg). Body temperature was maintained at 37°C throughout the experiment using a heating pad. Recordings were conducted in an acoustic chamber (MAC-1, Industrial Acoustic Company, Naperville, IL, USA). Mice were presented with click stimuli and pure tones at 6, 12, 18, 24, 30, and 35 khz, at intensities ranging from 10 to 90 dB-SPL, in steps of 5 dB. For each frequency–intensity combination, 512 responses were recorded and averaged. Responses were picked up by subdermal electrodes connected to a head-stage. ABR threshold was defined as the lowest sound intensity at which a reproducible waveform was observed. P1-N1 amplitudes and P1 latencies were extracted using a designated R algorithm (Rstudio, Boston, MA). For DPOAE recording, two speakers presenting two primary tones (f1 and f2) at a frequency ratio of 1.2 were coupled to a microphone and introduced into the ear canal of mice. Each frequency–tone combination was presented 256 times, and the results were averaged. The amplitude of the distortion product at a frequency of 2f1–f2 and the surrounding average noise level were extracted from the averaged responses using a designated R algorithm (Rstudio, Boston, MA). All measurements were performed using an RZ6 multiprocessor, MF1 speakers (Tucker-Davis Technologies, Alachua, FL), and an ER-10b+ microphone (Etymotic Research, Elk Grove Village, IL), and analyzed using BioSigRZ software (Tucker-Davis Technologies, Alachua, FL). All experiments were performed by the same tester.

### Cued fear conditioning

Mice were placed in a cage inside an acoustic chamber. After 90 s, a 20 s tone pip at 6 kHz was presented, followed by a 2 s 0.7 mA electric shock delivered through the cage. After an additional period of 60 s, the tone was presented again followed by an additional shock. The next day, the mice were placed in the same cage. After 150 s, the same tone appeared for 150 s, but was not followed by a shock. A camera was used to track the movement of the mouse continuously. The video was then divided into two time bins: 01:30–02:30 (before the tone appears) and 02:30–03:30 (after the tone appears). The video was then analyzed to evaluate the activity level and the degree of "freezing" behavior displayed by the mouse (EthoVision XT, Noldus) in each time bin. EthoVision XT uses objective parameters of pixel change in order to quantify activity.

Activity level is an automated measurement of the frame-to-frame change in pixels, reflecting the movement of the animal. Freezing behavior is defined as the lack of all movement, except that necessary for breathing. During this period, the activity level drops dramatically. We adjusted the threshold of freezing detection by manually inspecting videos of animals exhibiting freezing behavior and set a uniform threshold for all experiments. An example of this experiment is shown in Movie EV1 (red bars indicate detected freezing behavior, and red line indicates detected activity level). All experiments were carried out during daytime and by the same tester.

### Balance assessment

For the open-field test, mice were placed in a square 2.5 m$^2$ arena for 15 min and tracked using a camera. The video was then analyzed for rotation behavior, distance traveled, and cumulative duration spent in the center of the arena (EthoVision XT, Noldus). The software identifies the center point of the mouse and its nose in order to detect rotations. Both clockwise and counterclockwise rotations were counted. Distance traveled is a raw measurement of the pixel displacement in cm. The center of the arena was defined as the central 0.4 m$^2$ to evaluate the time when mice were not touching the walls of the arena. We analyzed the videos at time 05:00–15:00 to allow mice to adapt to their new environment. All experiments were carried out during the day and by the same tester.

### Immunofluorescence

Neonatal mice were sacrificed by decapitation, and adult mice were sacrificed by $CO_2$ inhalation. Inner ears were dissected under operating binoculars and fixed in 4% PFA for 2 h at room temp. After fixation, inner ears were washed in PBS and stored in 4°C until dissection. Inner ears of mice older than P10 were decalcified in 0.25 M EDTA until entirely soft. The organ of Corti was dissected in PBS under operating binoculars. Specimens were permeabilized and blocked in 2% Triton X-100 and 10% normal goat serum for 2 h at room temperature, and were then incubated overnight at 4°C in the appropriate primary antibody diluted in Phosphate Green antibody diluent (Bar Naor Ltd) according to manufacturer instructions. Specimens were washed and incubated for 2 h at room temperature in the relevant secondary antibody diluted in PBS according to manufacturer instructions and then mounted in ProLong Gold Antifade Mountant (Thermo Fisher Scientific) and imaged using a Zeiss LSM 880 (Zeiss, Oberkochen, Germany) equipped with an Airyscan detector. Antibody and stain concentrations were as follows: rabbit polyclonal myosin VIIa (Proteus Biosciences 25-6790) 1:250, mouse anti-FLAG (Sigma F3165) 1:1,000, DAPI (Abcam ab228549) 1:1,000, goat anti-mouse (Abcam ab150119) 1:250, and goat anti-rabbit Alexa Fluor 488 (Cell Signaling 4412s).

### FM1-43 uptake assay

P8 mice were sacrificed by $CO_2$ inhalation, and the sensory epithelium was dissected in PBS and cultured on Matrigel-coated Matek plates (In Vitro Technologies, Australia, FAL356237) for 24 h, as described previously (Goodyear et al, 2008). Growth medium was high-glucose DMEM (Biological Industries, Israel, 01-053-1A)

containing 1% fetal bovine serum (FBS) and 1% N2 supplement. FM 1-43 (Invitrogen, T3163) was diluted to 5 μM in PBS and applied to cochlear cultures for 10 s followed by three washes in PBS to prevent endocytic uptake. After 5 min, cultures were fixed in 4% PFA and imaged using Zeiss LSM 880.

### Image analysis

All data processing was performed off-line using commercial software packages (MATLAB R2019b, MathWorks Inc, Natick, MA, Ilastik and Fiji). For 3D surface projections, Imaris 8.4 software was used (Bitplane, Belfast, UK). A semi-automatic analysis code was used for cell detection in the organ of Corti and counting of inner and outer hair cells. To define position along the cochlea, a polyline (starting at the apex) was drawn along the pillar cell region and was segmented into 100 μm bins in which IHC and OHC were counted. For quantification of nuclei position, an apical to basal arc was fitted by manually marking the apical surface, nucleus center, and basal end of each HC. The code then fits an arc through the three positions in space and reports the distance of the nucleus from the apical surface according to voxel size. A custom FIJI macro was used to measure GFP intensity following viral transduction. IHC and OHC were manually identified based on myosin VIIa staining, and their nuclei were detected and segmented based on DAPI staining using automatic thresholding and Watershed. Mean GFP intensity was calculated in the area of the detected nuclei. An un-injected littermate was used as a control, and the mean fluorescence intensity measured in the nuclei of the control mouse was used to subtract the background. Plots show GFP intensity as arbitrary units normalized to the control. All mice (three injected with AAV.GFP and one control) were injected, dissected, and imaged on the same day using the same acquiring settings. IHC and OHC were regarded as positive if the measured GFP intensity was higher than 2 standard deviations above the mean fluorescence of the IHC and OHC in control mouse. For FLAG and DAPI intensity, a line was centered at the periphery of the nucleus, defined by the maximal DAPI fluorescence. Intensity was normalized to the maximal signal detected in each channel. All codes and macros are available upon request.

### Statistics

Statistical tests, group sizes, and *P* values are noted in the figure legends. Littermates were randomized to receive treatment or control. No blinding was performed, and all tests were carried out by the same tester. Objective measures were preferred when possible; these include hair cell counts, GFP and FLAG intensity, ABR P1-N1 amplitude and latency, DPOAE thresholds, and all behavioral outputs. Statistical analyses were performed using Prism 8 software (GraphPad, San Diego, CA). Correlation was computed using Pearson correlation. When required, Shapiro–Wilk and Kolmogorov–Smirnov were used to test data for normality. For comparisons of more than two groups or conditions, the Holm–Sidak post hoc test was used to adjust *P* values. Two animals were excluded from downstream analysis because they showed no improvement of auditory function. In one of them, immunofluorescence confirmed no transgene expression. We interpret this is technical injection failure. All *P* values are listed in Table EV1.

### The paper explained

#### Problem

Gene therapy is a promising strategy to treat genetic deafness. Since the auditory systems of humans and mice are very similar in structure, function, and even gene expression, mice serve as an excellent model for basic and translational auditory research. Genetic variants in *SYNE4*, encoding the nesprin-4 protein, have been shown to cause deafness in humans, and *Syne4*-deficient mice show a similar phenotype. In *Syne4*-knockout mice, the nuclei of outer hair cells (OHCs) lose their basal position and degenerate.

#### Results

We used a gene-replacement approach to rescue hearing in a mouse model of *SYNE4* deafness. This strategy required delivery of the coding sequence of *Syne4* into the inner ears of neonatal *Syne4*-knockout mice by a synthetic adeno-associated virus, AAV9-PHP.B. The results reveal near-complete rescue of hair cell morphology and survival, with normalization of auditory function and behavioral responses.

#### Impact

There are currently over 120 genes associated with inherited deafness. It is of paramount importance to test the feasibility of gene therapy in animal models in order to facilitate the development of future treatments. Our results provide proof of concept for the development of gene therapy for *SYNE4* and other forms of deafness.

### Study approval

All animal procedures were approved by the Animal Care and Use Committee (IACUC) at Tel Aviv University (01-17-101 and 01-19-084) and performed according to the NIH Guide for the Care and Use of Laboratory Animals.

# Data availability

This study includes no data deposited in external repositories.

**Expanded View** for this article is available online.

### Acknowledgements

The authors thank Wade Chien for training as we embarked on this project; Amy Patterson, Yukako Asai, Idan Schatz, Chris Walters (Tucker-Davis Technologies), Neil Ingham, and Tal Koffler for technical help and advice; and Hanna Grobe for assistance with image analysis. Behavioral tests were performed at the Myers Neuro-Behavioral Core Facility, Tel Aviv University with the assistance of Lior Bikoveski. The research was funded by the National Institutes of Health/NIDCD R01DC011835 (K.B.A.), the United States-Israel Binational Science Foundation (BSF) 01027150, Jerusalem, Israel (K.B.A., J.R.H.), Israel Science Foundation within the Israel Precision Medicine Partnership Program, 3499/19; the Capita Foundation (K.B.A., S.T.), the Jeff and Kimberly Barber Fund (J.R.H.) the Foundation Pour L'Audition (J.R.H.), and the European Research Council (ERC) under the European Union's Horizon 2020 Research and Innovation Programme, Grant Agreement No. 682161 (D.S.). K.B.A. is an incumbent of the Drs. Sarah and Felix Dumont Chair for Research of Hearing Disorders. This work was performed in partial fulfillment of the requirements for a Ph.D. degree by Shahar

Taiber, recipient of the Klass Family Fellowship, at the Sackler Faculty of Medicine, Tel Aviv University, Israel.

## Author contributions

KBA conceived the study. ST, KBA, JRH, and DS designed the study and interpreted the results. ST performed mouse injections, auditory testing, behavioral assays, cochlear dissections, and immunofluorescence and analyzed the data. OY-B cloned the AAV plasmids. RC wrote MATLAB codes for image analysis. ST, KBA, and JRH wrote the manuscript. All authors contributed to the article and approved the submitted version.

## Conflict of interest

J.R.H. holds a patent on use of AAV9-PHP.B for inner ear gene therapy and is an advisor to several biotech companies focused on inner ear therapeutics. The authors declare no other conflict of interests.

## For more information

https://www.nidcd.nih.gov/
https://hereditaryhearingloss.org
https://deafnessvariationdatabase.org
https://www.gtexportal.org
http://umgear.org
https://www.kbalab.com/
https://www.holtgeleoclab.com

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
