## [Review Process File · EMBO Molecular Medicine]

Neonatal AAV gene therapy rescues hearing in a mouse model of SYNE4 deafness

Shahar Taiber, Roie Cohen, Ofer Yizhar-Barnea, David Sprinzak, Jeffrey Holt, and Karen Avraham
DOI: [10.15252/emmm.202013259](https://doi.org/10.15252/emmm.202013259)

Corresponding author: Karen Avraham (karena@tauex.tau.ac.il)

Review Timeline:

Submission Date:	8th Aug 20
Editorial Decision:	1st Sep 20
Revision Received:	2nd Oct 20
Editorial Decision:	6th Nov 20
Revision Received:	10th Nov 20
Accepted:	18th Nov 20

Editor: Jingyi Hou

Transaction Report:

1st Sep 2020

Dear Karen,

Thank you for the submission of your manuscript to EMBO Molecular Medicine. We have now received feedback from the three referees whom we asked to evaluate your manuscript. As you will see from the reports below, the referees acknowledge the interest of the study. However, they also raise a series of concerns about your work, which should be convincingly addressed in a major revision of the present manuscript.

I think that the referees' recommendations are rather clear and there is no need to reiterate their comments. In particular, the referees mentioned that this manuscript can benefit from adding additional data on AAV delivery at later developmental stages, and they pointed out that the direct role of Syne4 in rescuing central auditory functions remains rather unclear at this stage, which we would strongly encourage you to address.

All other issues need to be addressed as well. We would welcome the submission of a revised version within three months for further consideration. Please note that EMBO Molecular Medicine strongly supports a single round of revision and that, as acceptance or rejection of the manuscript will depend on another round of review, your responses should be as complete as possible.

We are aware that many laboratories cannot function at full efficiency during the current COVID-19/SARS-CoV-2 pandemic and have therefore extended our "scooping protection policy" to cover the period required for a full revision to address the experimental issues. Please let me know should you need additional time, and also if you see a paper with related content published elsewhere.

I look forward to receiving your revised manuscript.

Sincerely,
Jingyi

Jingyi Hou, PhD
Editor
EMBO Molecular Medicine

*** Instructions to submit your revised manuscript ***

** PLEASE NOTE ** As part of the EMBO Publications transparent editorial process initiative (see our Editorial at <https://www.embopress.org/doi/pdf/10.1002/emmm.201000094>), EMBO Molecular Medicine will publish online a Review Process File to accompany accepted manuscripts.

To submit your manuscript, please follow this link:

Link Not Available

- 1) a .docx formatted version of the manuscript text (including Figure legends and tables). Please make sure that the changes are highlighted to be clearly visible to referees and editors alike.
- 2) separate figure files*
- 3) supplemental information as Expanded View and/or Appendix. Please carefully check the authors guidelines for formatting Expanded view and Appendix figures and tables at <https://www.embopress.org/page/journal/17574684/authorguide#expandedview>
- 4) a letter INCLUDING the reviewers' reports and your detailed responses to their comments (as Word file)

Also, and to save some time should your paper be accepted, please read below for additional information regarding some features of our research articles:

- 5) The paper explained: EMBO Molecular Medicine articles are accompanied by a summary of the articles to emphasize the major findings in the paper and their medical implications for the non-specialist reader. Please provide a draft summary of your article highlighting
 - the medical issue you are addressing,
 - the results obtained and
 - their clinical impact.

- 6) For more information: There is space at the end of each article to list relevant web links for further consultation by our readers. Could you identify some relevant ones and provide such information as well? Some examples are patient associations, relevant databases,

OMIM/proteins/genes links, author's websites, etc...

7) Author contributions: the contribution of every author must be detailed in a separate section (before the acknowledgments).

8) EMBO Molecular Medicine now requires a complete author checklist (<https://www.embopress.org/page/journal/17574684/authorguide>) to be submitted with all revised manuscripts. Please use the checklist as a guideline for the sort of information we need WITHIN the manuscript as well as in the checklist. This is particularly important for animal reporting, antibody dilutions (missing) and exact p-values and n that should be indicated instead of a range.

9) Every published paper now includes a 'Synopsis' to further enhance discoverability. Synopses are displayed on the journal webpage and are freely accessible to all readers. They include a short stand first (maximum of 300 characters, including space) as well as 2-5 one sentence bullet points that summarise the paper. Please write the bullet points to summarise the key NEW findings. They should be designed to be complementary to the abstract - i.e. not repeat the same text. We encourage inclusion of key acronyms and quantitative information (maximum of 30 words / bullet point). Please use the passive voice. Please attach these in a separate file or send them by email, we will incorporate them accordingly.

You are also welcome to suggest a striking image or visual abstract to illustrate your article. If you do please provide a jpeg file 550 px-wide x 400-px high.

10) A Conflict of Interest statement should be provided in the main text

11) Please note that we now mandate that all corresponding authors list an ORCID digital identifier. This takes <90 seconds to complete. We encourage all authors to supply an ORCID identifier, which will be linked to their name for unambiguous name identification.

Currently, our records indicate that the ORCID for your account is 0000-0002-4913-251X.

Link Not Available

12) The system will prompt you to fill in your funding and payment information. This will allow Wiley to send you a quote for the article processing charge (APC) in case of acceptance. This quote takes into account any reduction or fee waivers that you may be eligible for. Authors do not need to pay any fees before their manuscript is accepted and transferred to our publisher.

Photos 400-800 DPI

Figures are not edited by the production team. All lettering should be the same size and style; figure

panels should be indicated by capital letters (A, B, C etc). Gridlines are not allowed except for log plots. Figures should be numbered in the order of their appearance in the text with Arabic numerals. Each Figure must have a separate legend and a caption is needed for each panel.

*Additional important information regarding figures and illustrations can be found at <http://bit.ly/EMBOPressFigurePreparationGuideline>

***** Reviewer's comments *****

Referee #1 (Comments on Novelty/Model System for Author):

I think this is a nice study and the data seem of be of high quality and reliable. While inner ear gene therapy has been successfully applied to several mouse models of hereditary hearing loss, this is the first report of successful application of inner ear gene therapy to the *Syne4*^{-/-} mouse.

Referee #1 (Remarks for Author):

In this study, the authors tested whether inner ear gene therapy could be used to rescue hearing in the *Syne4*^{-/-} mouse. They found that AAV9-PHP.B-*Syne4* was able to rescue hair cell morphology and survival, as well as near complete restoration of auditory function at 1 month. I think this is a nice study which showed that gene therapy is effective at improving auditory function in a mouse model of hearing loss. My comments are listed below:

1. The authors stated that nearly all IHCs and OHCs were infected by AAV9-PHP.B-GFP. Yet, in figure 3, it seems that the infection is mostly in the OHCs, and not so much in the IHCs (Figure 3B). This could be an imaging level issue. In Figure 3C, it seems that not many HCs (which are *Myo7a* positive)also co-express GFP. Perhaps orthogonal sections would be more convincing to show the HC transduction rate.
2. Figure 3D, it is not clear to me how GFP intensity is quantified. I would suggest a more detailed explanation on how this is done in the methods section. How many mice were used for this experiment?
3. It is interesting that the effect of auditory improvement with gene therapy decreased by 12 weeks. It seems that IHCs and OHCs are present in the mutant mice after gene therapy treatment. Does the position of the nuclei change overtime?
4. The authors stated that *Syne4* is expressed in the central nervous system. They used a cued-fear conditioning test to assess whether inner ear gene therapy can "rescue central auditory function". I am not sure the data presented in this study showed any "rescuing" of central auditory function. Was *Syne4* expression restored in the central auditory system? If so, the data was not shown in the manuscript. Isn't it likely that the presence of the cued-fear response in the treated mutant mice is due to the recovery of peripheral auditory function in the inner ear?
5. Similarly, in the discussion, the authors wrote, "it was interesting to examine whether the observed physiological recovery of the inner ear and auditory brainstem could drive complex behavioral responses that rely on comprehensive central processing." Again, I don't think the data presented in this study showed any recovery in the auditory brainstem.

Referee #2 (Remarks for Author):

This is an elegant and important paper showing that AAV9-PHP.B gene therapy given to immature

ears of mice modeling DFNB76 leads to rescue of structure and function in these ears.

The work is original and the presentation is clear.

There are several critical issues that need to be addressed, and some other minor points that could improve the paper once resolved.

Critical to fix:

Treatment was only given at one very early age, when the hair cells are in early developmental stage. Much clearer mention of this fact is needed, ideally, in the title, abstract, and discussion.

Discussion is needed for the possibility that the results are dependent on age. Was the injection attempted at a later stage in development while OHCs are still alive? If so, adding the data would make the story more complete and useful.

It would also be useful to include better documentation of the onset of AAV9 gene expression, and make the point that by the time transgene expression start, hair cells are further along their development.

Adding later age injections would also increase the novelty, as most other phenotypic rescue studies published to date are using early developmental stage interventions.

Please also include susceptibility of the relevant cells to transfection and normal timing of expression of the gene that is being inserted. The latter seems especially important in light of the relatively late appearance of hearing loss in the target patient population.

The images in figure 3 are clear and crisp (nice!) and do not correspond well with the quantitation.

Based on these clear images it looks like most of the 1st row outers and inner pillars are gfp-negative, and it looks like inner phalangeal cells, not IHC, are weakly positive. All IHCs are negative.

This is a complete mis-match with the counting, and difficult to explain along with the change in nuclear position in the IHCs. It is also surprising considering that most AAVs tried in the inner ear so far transduced IHCs with higher efficiency than any other cell type.

There are several issues with the behavioral tests (or their presentation). The basis of the shock test is not clear in the methods section (freezing rate before subtracted from after? Paired t-test? Repeated measures test with time bins before and after sound presentation?). The figure caption indicates a 2-way ANOVA was used, but the result is only given for one factor, before vs after within treatment groups, the between groups result is not given. A one-way MANOVA or repeated measures test (including pairwise t-test) would be better to account for the possibility that behaviors before and after sound presentation could be different. Freezing rate and duration could both be indicators of fear, both could be elevated after return to the test cage because the shock would be associated with that cage and both could change after sound presentation. There is even less detail about evaluation of activity patterns.

The issues related to using the two-way ANOVA also apply to the ABR and DPOAE analyses.

Sample sizes are small for some subgroups raising questions about consistency. Plots show substantial variation among individuals in the better represented subgroups, raising questions about reproducibility (i.e., some p-values could reflect sampling effects and not be representative of the general tendency).

Other points

Some of the citations are not carefully selected. For instance, the Raphael and Altschuler citation is not a suitable one (even when the cited author is the reviewer here). Please go over the entire list and ascertain whether the most appropriate citations are chosen.

The introduction is too long and includes parts that read like a review or a chapter about inner ear structure and function. It is better to focus on the main topic and to send readers who are not ear experts to reviews elsewhere.

In the sentence: "with a focus on the organ of Corti and hair cells", delete "and hair cells"

What the images in 4c show is difficult to figure out. Numbers need to be larger and a clear, detailed explanation included.

Balance assessment methods need more details, especially for time in center (how was center defined?).

In fear condition methods, what is the criterion for "freezing"? The results need revision to be clear that freezing in these windows were tabulated and compared. What statistical test was used and how were these two counts treated in that test?

...cleaned from iodine.... use 'with' if you mean iodine and ethanol were used to perform the cleaning, or use 'of' if you mean that residual iodine and ethanol were removed from the pups.

Not sure what the following sentence means and how it enhances the importance of the work:

..."many of the deafness genes affect only a handful of families. Collectively, these account for a substantial proportion of patients with deafness. While a translational gap exists for treatment of rare diseases (Tambuyzer et al, 2020)..."

Referee #3 (Comments on Novelty/Model System for Author):

The mouse is an excellent model for studying inner ear defects and deafness in humans.

Referee #3 (Remarks for Author):

Hearing loss is the most frequent sensory disorder (466 millions worldwide), with half the cases of congenital and early-onset deafness due to genetic causes. Deafness is extremely heterogeneous, with over 120 genes associated with non-syndromic hearing loss in humans. Pathogenic variants in SYNE4, encoding nesprin-4 (a member of the Linker of Nucleoskeleton and Cytoskeleton (LINC) complex), have been found to cause autosomal recessive progressive DFNB76 hearing loss in humans. In this report, S. Taiber and colleagues used *Syne4*^{-/-} mice that display severe-to-profound progressive hearing loss that is reminiscent of DFNB76 and ascribed to mislocalization of hair cell nuclei and hair cell degeneration. Hair cell counts and FM1-43 staining reveal almost normal hearing organ by P8, with readily apparent degeneration of the outer hair cells (OHC) by P14, at the onset of hearing. *Syne4*^{-/-} mice exhibited no abnormal balance behavior. To test the therapeutic effect of *Syne4* delivery, the authors used neonatal injections of a synthetic AAV with high transduction rates to inner and outer hair cells, the AAV9-PHP.B vector. ABR and DPOAEs measurements indicated nearly complete recovery of auditory function at 4 and 8 weeks of age in treated ears. Measurements at 12 weeks showed a decrease of the hearing recovery, but still. Also, hair cells count confirm survival of virtually all OHCs in treated *Syne4*^{-/-} mice, lasting up to 12 weeks post injection. Nuclei of OHCs from injected ears were positioned at the basal part of the cell, as in wild-type mice.

This work shows that AAV therapy can apply to the outer hair cells, highly specialized auditory sensory cells so far refractory to transduction by multiple AAVs. The rates of transduction obtained in cochlear hair cells are important achievements and the successful gene therapy is of interest to a broader audience and very relevant to the research community focused on the study of genes associated with deafness phenotype.

There are a number of points the authors could answer to improve further the manuscript

1- FM1-43 staining fail to reveal IHC cells activity. It is unclear why the authors chose to assess the hair cell mechanotransduction in organotypic maintained in culture. Either recording the mechanotransduction *ex vivo*, or FM1-43 on fresh organs should be more relevant. If mechanotransduction is affected beyond P14 in *Syne4*^{-/-} mice, FM1-43 can be used to monitor

potential rescue after AAV therapy.

2- It wasn't clear either why the authors used the PSC route for gene delivery to target auditory hair cells? Did the authors have data with injections through round window membrane? Any information on AAV delivery at later stages would provide key information on therapeutic window.

3- While the experiment in Fig 7 confirms perception followed by action after sound processing, it's not clear if Syne4 has a direct critical role in the central auditory pathways? To make Syne4 function in central pathways a critical finding of the paper, the causal versus correlation between loss of Syne4 and effect on central processing need be clarified. Experiments in young Syne4^{-/-} mice using sound stimuli at frequencies and high intensities to overcome the moderate hearing loss at early stages can help bypass the peripheral sensitivity in absence of Syne4.

4- Page 9: lines 1-2: In sentence "Another mouse showed no recovery and was excluded from the DPOAE analysis." Why this mouse showed no recovery and why was it excluded?

5- Page 3: in sentence "... proteins such as kinesins and dynemin..." : "dynein" instead of "dynemin"

6- Fig.1 is unnecessary as main Figure. If needed, Fig1B can be added to current Figure 2.

7- Fig. 3F: to illustrate further the localization of Syne4 after AAV.Syne4 delivery, I encourage the authors to provide a high magnification image showing protein immunolocalization on sagittal sections of OHCs (e.g. similar to OHCs in Fig 4B). Is Flag.Syne4 also expressed in nuclear membranes of vestibular hair cells?

8- For sake of clarity, "AAV9" instead of "AAV" should be used throughout, including in figures. Also, replace "Mut" with "Syne4^{-/-}"; there is a mixed use of WT or wild-type in legends: homogenize throughout the text;

Referee #1

1. The authors stated that nearly all IHCs and OHCs were infected by AAV9-PHP.B-GFP. Yet, in figure 3, it seems that the infection is mostly in the OHCs, and not so much in the IHCs (Figure 3B). This could be an imaging level issue. In Figure 3C, it seems that not many HCs (which are Myo7a positive) also co-express GFP. Perhaps orthogonal sections would be more convincing to show the HC transduction rate.

Response: This is an excellent remark that prompted us to revisit the data and produce a clearer figure. We now show the IHC and OHC separately, each in the plane that shows GFP most clearly. We also included YZ projections to show the distribution of GFP in the HC.

2. Figure 3D, it is not clear to me how GFP intensity is quantified. I would suggest a more detailed explanation on how this is done in the methods section. How many mice were used for this experiment?

Response: Three injected mice and one uninjected control were used for this experiment. We have revised the results, figure legend, and methods to clearly state this. We have now expanded the section in the methods describing the quantification.

In the Results:

The normalized GFP intensity was similar between three individual injected mice and typically higher in OHC as compared to IHC (Fig. 3D)

In the figure legend:

Top shows OHC plane, bottom shows IHC plane, right shows YZ orthogonal projection. Black asterixis show bright Deiters cells.

In the Methods:

In order to measure GFP intensity following viral transduction, a custom FIJI macro was used. IHC and OHC were manually identified based on Myosin VIIa staining and their nuclei were detected and segmented based on DAPI staining using automatic thresholding, and Watershed. Mean GFP intensity was calculated in the area of the detected nuclei. An uninjected littermate was used for control, and the mean fluorescence intensity measured in the nuclei of the control mouse was used to subtract the background. Plots show GFP intensity as arbitrary units normalized to the control. All mice (three injected with AAV.GFP and one control) were injected, dissected, and imaged on the same day using the same acquiring settings. IHC and OHC were regarded as positive if the measured GFP intensity was higher than 2 standard deviations above the mean fluorescence of the IHC and OHC in control mouse. All codes and macros are available upon request.

3. It is interesting that the effect of auditory improvement with gene therapy decreased by 12 weeks. It seems that IHCs and OHCs are present in the mutant mice after gene therapy treatment. Does the position of the nuclei change overtime?

Response: We thank the reviewer for this important comment. We have now analyzed OHC nuclei position at 12w in WT mice and *Syne4*^{-/-} mice injected with AAV.Syne4 (Fig 6 E-G). We could not detect any significant change in nuclei position. We cannot provide a supported mechanism by which auditory function deteriorates over time in the treated mice. An immune response against the AAV capsid, or toxicity caused by overexpression of *Syne4* are not likely to be the cause of the observed deterioration since WT mice injected with the virus show no such decrease in auditory function. We speculate that either *Syne4* expression is important for other parts of the auditory system that we are not aware of, or that an immune response against the transgene, which does not occur in WT mice since they have endogenous expression of *Syne4*, leads to some form of damage that is not detectable with methods used in this manuscript.

The following text has been added in the Results section:

We tested whether the observed deterioration of auditory function in some mice in the treatment group could be explained by a change nuclei position, one that does not lead to OHC death but impairs their function. For this purpose, we quantified the position of OHC nuclei at the 12 kHz region of the organ of Corti at 12 weeks (Fig 6 E-G). We could not detect a significant change in their position at 12 weeks, suggesting that this could not explain the deterioration we observed in some of the treated mice.

In the Discussion:

Nuclei of OHCs from injected ears were positioned at the basal part of the cell, as in wild-type mice, and there was no observed difference in long-term OHC survival or nuclei position... We are not sure whether this is due to silencing of the transgene, the result of an immune reaction against the viral capsid or the transgene or must be attributed to another cause. We did not observe substantial loss of OHC at 12 weeks, neither did we observe any significant change in nuclei position that could explain why some mice performed worse than others.

4. The authors stated that *Syne4* is expressed in the central nervous system. They used a cued-fear conditioning test to assess whether inner ear gene therapy can "rescue central auditory function". I am not sure the data presented in this study showed any "rescuing" of central auditory function. Was *Syne4* expression restored in the central auditory system? If so, the data was not shown in the manuscript. Isn't it likely that the presence of the cued-fear response in the treated mutant mice is due to the recovery of peripheral auditory function in the inner ear?

Response: Yes, in fact, that is exactly what we meant. Our description of this experiment and the interpretation of its results were not clear enough. Because *Syne4* expression was detected in the

central nervous system, we wished to test whether our treatment, that is applied locally to the inner ear, can rescue behaviors that are dependent on comprehensive auditory processing. Cued-fear conditioning is classical assay in which freezing behavior has been demonstrated to rely on components of auditory system in the CNS, such as the medial-geniculate body (Weinberger, 2011; Courtin et al, 2013; Ciochi et al, 2010). If Syne4 expression was critical for the function of auditory pathways in the CNS, we would expect our treatment to successfully rescue peripheral auditory function (evaluated by ABR and DPOAE) but not central auditory processing that is necessary to drive behaviors such as cued-fear conditioning. If these results reflect the pathology in humans with Syne4 deafness, we reason that peripheral delivery of SYNE4 should be sufficient to rescue auditory function. We have now edited the results and discussion to convey these ideas more clearly:

In the Results:

...Syne4 is expressed in parts of the central nervous system (GTEX Portal). Therefore, Syne4 could have a role in the function of the central auditory system and peripheral delivery of Syne4 might not rescue auditory functions that rely on central processing of sound... investigate whether peripheral delivery of exogenous Syne4 into Syne4^{-/-} mice would be sufficient to also rescue auditory behaviors that require central auditory processing...

In the Discussion:

Although Syne4 is expressed in the CNS, it is not clear whether Syne4 is necessary for central auditory function, as is the case for some deafness genes... Our results do not support the possibility of a central role for Syne4, but rather suggest that it is only necessary for normal organ of Corti function.

5. Similarly, in the discussion, the authors wrote, "it was interesting to examine whether the observed physiological recovery of the inner ear and auditory brainstem could drive complex behavioral responses that rely on comprehensive central processing." Again, I don't think the data presented in this study showed any recovery in the auditory brainstem.

Response: In our study we have used auditory brainstem response (ABR), distortion-product otoacoustic emissions (DPOAE), and cued-fear conditioning to assess auditory function. The five peaks observed in mouse ABR correspond to the cochlear nerve, superior olivary complex, lateral lemniscus, and inferior colliculus (Willot, 2006). Therefore, this test encompassed the entirety of the canonical auditory pathway in the brainstem.

Referee #2

Critical to fix:

Treatment was only given at one very early age, when the hair cells are in early developmental stage. Much clearer mention of this fact is needed, ideally, in the title, abstract, and discussion. Discussion is needed for the possibility that the results are dependent on age. Was the injection attempted at a later stage in development while OHCs are still alive? If so, adding the data would make the story more complete and useful.

Response: We thank the reviewer for this comment. Multiple attempts have been made to deliver viral vectors at later stages. While this has been achieved by other before, unfortunately, we were not able to obtain reliable data from these experiments. High mortality rates of mice due to anesthesia, poor recovery, and high variability in transduction rates, have prevented us from assessing the potential of later treatment. While Syne4^{-/-} mice are born with all OHC in place, by the age of P14 the majority of them are already gone. Therefore, we believe that even if later delivery would succeed, there would only be a few more days after birth when treatment would be relevant. Importantly, as we

note in the discussion, the phenotype observed in humans with SYNE4 deafness is much more gradual, so we hope that a wider therapeutic time window exists for these patients.

We have now edited the text in multiple locations to emphasize this point:

Title:

Neonatal AAV gene therapy rescues hearing in a mouse model of SYNE4 deafness

Abstract:

... deliver the coding sequence of *Syne4* into the inner ears of neonatal *Syne4*^{-/-} mice.

Discussion:

It is not clear whether a later intervention would still be relevant, but since the majority of OHC in these mice rapidly degenerate at the onset of hearing, and since several days pass between injection and expression of the transgene (Lee et al., 2020), we believe that the time window may be restricted... Our results indicate that exogenous delivery of *Syne4* into neonatal *Syne4*^{-/-} hair cells...

It would also be useful to include better documentation of the onset of AAV9 gene expression, and make the point that by the time transgene expression start, hair cells are further along their development.

Response: We have characterized the onset of expression for AAV9-PHP.B before (Lee et al, 2020, Fig 5B). We have now revised the text to address this issue:

Results:

Expression of GFP delivered in AAV9-PHP.B begins rising reliably between day 3 and 5 post injection (Lee et al., 2020)

Discussion:

... since several days pass between injection and expression of the transgene (Lee et al, 2020).

Adding later age injections would also increase the novelty, as most other phenotypic rescue studies published to date are using early developmental stage interventions.

Response: We have addressed the issue of delivery in the previous comment.

Please also include susceptibility of the relevant cells to transfection and normal timing of expression of the gene that is being inserted. The latter seems especially important in light of the relatively late appearance of hearing loss in the target patient population.

Response: This is an excellent point risen by the reviewer. Relevant data in humans is not available, but previous studies in mice indicate that low levels nesprin 4 (the protein product of *Syne4*) can be detected in OHC already at P0, with a clear staining at P12 (Horn et al, 2013). Exploration of RNAseq data shows that *Syne4* is not detected at embryonic stages, but at the RNA level is already detected at P0 (et al). Despite several attempts, we could not achieve specific staining for nesprin 4 using commercial and in-house antibodies. We hope to achieve a better understanding of the exact timing of nesprin 4 activity in organ of Corti in the future.

We have now revised Fig EV2 to include developmental RNAseq data from Scheffer et al, 2015 available via umgear.org and revised the text to address this issue:

Results:

Syne4 RNA is detected as early as P0, although the staining at P0 is weaker than at P12 (Horn et al, 2013, Fig EV2B)

The images in figure 3 are clear and crisp (nice!) and do not correspond well with the quantitation. Based on these clear images it looks like most of the 1st row outers and inner pillars are gfp-negative, and it looks like inner phalangeal cells, not IHC, are weakly positive. All IHCs are negative. This is a complete mis-match with the counting, and difficult to explain along with the change in nuclear position in the IHCs. It is also surprising considering that most AAVs tried in the inner ear so far transduced IHCs with higher efficiency than any other cell type.

Response: In response to a similar comment by another reviewer (comments number 1 and 2 by referee #1) we have revised the Fig 3 to include 2 separate focal planes in order to better visualize OHC and IHC, as well as a YZ orthogonal projection. We have also added more elaborate explanation of how GFP intensity was quantified. We hope these changes will help the readers and the reviewer see the GFP signal in the relevant cells.

The observation that OHC show higher fluorescence than IHC is indeed surprising. We believe it could be explained either by the delivery route used in this study, or the construct we used that contains turboGFP as opposed to GFP that is standardly used.

There are several issues with the behavioral tests (or their presentation). The basis of the shock test is not clear in the methods section (freezing rate before subtracted from after? Paired t-test? Repeated measures test with time bins before and after sound presentation?). The figure caption indicates a 2-way ANOVA was used, but the result is only given for one factor, before vs after within treatment groups, the between groups result is not given. A one-way MANOVA or repeated measures test (including pairwise t-test) would be better to account for the possibility that behaviors before and after sound presentation could be different. Freezing rate and duration could both be indicators of fear, both could be elevated after return to the test cage because the shock would be associated with that cage and both could change after sound presentation. There is even less detail about evaluation of activity patterns.

Response: We thank the reviewer for this comment. The mice indeed show freezing behavior even without any tone. This is good because it means that memory is formed properly, even without Syne4. However, only wild-type mice and treated Syne4^{-/-} mice show significantly more freezing after the tone appears compared to before. This indicates that they respond to the tone, while untreated Syne4^{-/-} mice and Syne4^{-/-} treated with AAV.GFP do not. The activity level is an automated measurement that does not require any thresholding and shows a mirror image of freezing behavior: when mice can hear the tone the activity level drops significantly. This is described in the results: "Since the mice could not hear the stimulus, they probably associated the shock with other clues found in the scene, such as the smells and appearances of the cage, the room, and the tester", and in the discussion: "This indicates that memory was formed but could not be associated with the stimulus."

An example of how this experiment is performed and analyzed is provided in EV movie 1. We bring here two snapshots of a mouse during movement (a) and during freezing (b). purple pixels indicate frame-to-frame change.

(a)

(b)

We have changed the statistical test to Repeated Measures ANOVA, comparing the score of each mouse in the 2 time bins (no tone vs with tone). We have now revised the text to explain how these experiments were done and how the data were analyzed.

In the Methods section:

The video was then divided into two time bins: 01:30-02:30 (before the tone appears), and 02:30-03:30 (after the tone appears). The video was then analyzed to evaluate the activity level and the degree of “freezing” behavior displayed by the mouse (EthoVision XT, Noldus) in each time bin. EthoVision XT uses objective parameters of pixel change in order to quantify activity. Activity level is an automated measurement of the frame-to-frame change in pixels, reflecting the movement of the animal. Freezing behavior is defined as the lack of all movement, except that necessary for breathing. During this period, the activity level drops dramatically. We adjusted the threshold of freezing detection by manually inspecting videos of animals exhibiting freezing behavior and set a uniform threshold for all experiments. An example of this experiment is shown in EV Movie 1.

In figure legend:

Statistical test was Repeated-Measures ANOVA with Holm-Sidak correction for multiple comparisons, comparing the scores of each individual mouse before and after the appearance of the tone.

The issues related to using the two-way ANOVA also apply to the ABR and DPOAE analyses.

Response: In our ABR and DPOAE experiments we are comparing several groups (WT, $Syne4^{-/-}$, $Syne4^{-/-}$ treated with AAV.Syne4 and $Syne4^{-/-}$ treated with AAV.GFP). Additionally, each subject is measured several times (several frequencies). The data passed both the Shapiro-Wilk test and the Kolmogorov-Smirnov tests for normality. The only exception we observed was the $Syne4^{-/-}$ group in the 12-week ABR data that did not pass the Shapiro-Wilk test but did pass the Kolmogorov-Smirnov test. Therefore, ANOVA was chosen as the appropriate test in both cases.

Sample sizes are small for some subgroups raising questions about consistency. Plots show substantial variation among individuals in the better represented subgroups, raising questions about reproducibility (i.e., some p-values could reflect sampling effects and not be representative of the general tendency).

Response: In order to ensure the reproducibility of our data and to avoid potential biases as much as possible all mice were randomly assigned to the different treatment groups. In addition, all physiological measurements, analyses, and behavioral tests were performed by the same tester. We could not blind the tester as the animals had to be marked for the follow-up experiments, but we chose objective, quantitative measurements when possible (HC counts, GFP and FLAG intensity, ABR P1-N1

amplitude and latency, DPOAE thresholds, and all behavioral outputs were quantified in an automated manner).

In order to make sure we obtain a good representation of the different groups the following group sizes were tested: For FM 1-43 uptake each group consisted of 3 mice and showed the same qualitative result. For transduction efficiency experiments 3 littermates were injected on the same day and one littermate served as control. We observe variability between different cells in the same cochlea but no difference in the overall intensity levels and/or transduction rates. For FLAG intensity measurements 8 individual cells were measured. For OHC nuclei position at 4 weeks we measured 30-45 IHC in each group and 81-135 OHC for each group. For ABR tests performed at 4 weeks we tested 6-21 mice in each group for thresholds, the less represented groups are the *Syne4*^{-/-} mice and the *Syne4*^{-/-} mice treated with AAV.GFP. Since these mice are nearly completely deaf we do not see the need to test larger numbers. Importantly, the treatment group (*Syne4*^{-/-} mice treated with AAV.*Syne4*) requires much better representation because of the technical variability in the delivery of the AAV. Therefore, this group consisted of 20 mice. For P1-N1 amplitude and latency we show data for 4-7 mice but these data hardly show any variability. For DPOAE thresholds at 4 weeks we used 5-6 mice in each group. Again, these data show minor variability so we believe larger groups are not needed. For ABR tests performed at 12 weeks we show data for 5-16 mice in each group. In these data the treatment group, which consists of 16 mice, indeed shows substantial variability. Since we did not observe this amount of variability at 4 weeks with an even larger group, we believe this variability reflects the biological variability in the response and durability of the treatment rather than measurement or sampling issues. For long term survival of OHC we used 3-6 mice in each. Here as well, the treatment group is better represented for the reasons mentioned above. For OHC nuclei position at 12 weeks were measured 30-38 cells in each group. For fear conditioning we tested 5-16 mice in each group. In this case, we tested 16 *Syne4*^{-/-} to make sure that we reject the possibility of auditory-associated freezing behavior in these mice with sufficient statistical power. For ABR and DPOAE tests performed to characterize the safety of the treatment we show data for 5-10 mice in each group. For open field we show data for 7-9 mice in each group. Finally, for weight gain tracking we show data for 5-11 mice in each group.

These group sizes are consistent with previous studies in this field: Akil et al, 2019, Isgrig et al, 2017. We hope this answers the concerns raised by the reviewer regarding reproducibility and explains the rationale of the different group sizes.

Other points

Some of the citations are not carefully selected. For instance, the Raphael and Altschuler citation is not a suitable one (even when the cited author is the reviewer here). Please go over the entire list and ascertain whether the most appropriate citations are chosen.

Response: We removed the mentioned citation and revisited the citations throughout the manuscript to make sure they are suitable.

The introduction is too long and includes parts that read like a review or a chapter about inner ear structure and function. It is better to focus on the main topic and to send readers who are not ear experts to reviews elsewhere.

Response: We have now removed the paragraph describing the inner ear. The introduction is now 560 words

In the sentence: "with a focus on the organ of Corti and hair cells", delete "and hair cells"

Response: Deleted

What the images in 4c show is difficult to figure out. Numbers need to be larger and a clear, detailed explanation included.

Response: We have now edited Fig 4C to only show OHC so that the images are larger and clearer, we have made the numbers larger, and added description to the methods section and figure legend:

In the Methods:

The code then fits an arc through the 3 positions in space and reports the distance of the nucleus from the apical surface according to voxel size... All codes and macros are available upon request

In the figure legend:

Image analysis of nucleus position of OHC. Images show an arc fitted through the apical surface, nucleus, and basal end of the cell in 3D. X, Y, and Z axes show position in μm .

Balance assessment methods need more details, especially for time in center (how was center defined?).

Response: We thank the reviewer for this important comment. We have added description to this section and it now reads as follows:

For the open-field test, mice were placed in a square 2.5 m² arena for 15 mins and tracked using a camera. The video was then analyzed for rotation behavior, distance traveled, and cumulative duration spent in the center of the arena (EthoVision XT, Noldus). The software identifies the center point of the mouse and its nose in order to detect of rotations. Both clockwise and counterclockwise rotations were counted. Distance traveled is a raw measurement of the pixel displacement in cm. The center of the arena was defined as the central 0.4 m² to evaluate the time when mice were not touching the walls of the arena. We analyzed the videos at time 05:00-15:00 to allow mice to adapt to their new environment. All experiments were carried out during the day and by the same tester.

In fear condition methods, what is the criterion for "freezing"? The results need revision to be clear that freezing in these windows were tabulated and compared. What statistical test was used and how were these two counts treated in that test?

Response: This is an important comment raised by the reviewer and we have addressed it in a similar comment raised by this reviewer above ("There are several issues with the behavioral tests...").

...cleaned from iodine.... use 'with' if you mean iodine and ethanol were used to perform the cleaning, or use 'of' if you mean that residual iodine and ethanol were removed from the pups.

Response: The sentence has been corrected and now reads "cleaned of remaining iodine and ethanol".

Not sure what the following sentence means and how it enhances the importance of the work:

..."many of the deafness genes affect only a handful of families. Collectively, these account for a substantial proportion of patients with deafness. While a translational gap exists for treatment of rare diseases (Tambuyzer et al, 2020)..."

Response: We wish to emphasize that despite the fact the SYNE4 deafness is rare, it is still important to develop treatments for it. We have revised this paragraph and it now reads as follows: There are currently over 120 genes associated with non-syndromic hearing loss in humans. While some variants are prevalent in certain populations, such as GJB2 167delT in Ashkenazi Jews (Sobe et al, 1999) and TMC1 p.Ser647Phe in Moroccan Jews (Brownstein et al, 2011), many of the deafness genes affect only a handful of families. This raises concerns regarding the justification and feasibility of developing

personalized treatments for these forms of deafness. However, collectively, rare variants account for a substantial proportion of patients with deafness. To more broadly address the needs of hearing loss patients, therapeutic solutions will also be required for the less prevalent variants. While a translational gap exists for treatment of rare diseases (Tambuyzer et al, 2020), given the extent of recovery we observed in this study, we believe that gene therapy for SYNE4 deafness is not only feasible, but also imperative.

Referee #3

1- FM1-43 staining fail to reveal IHC cells activity. It is unclear why the authors chose to assess the hair cell mechanotransduction in organotypic maintained in culture. Either recording the mechanotransduction ex vivo, or FM1-43 on fresh organs should be more relevant. If mechanotransduction is affected beyond P14 in *Syne4*^{-/-} mice, FM1-43 can be used to monitor potential rescue after AAV therapy.

Response: In fact, IHC were also stained for FM1-43. We have now split the images to show different Z planes for OHC and IHC so that this is clearer. The FM1-43 assay is delicate. If not used carefully, it can lead to false positives. Through years of experience with this assay (Geleoc and Holt, 2003), we have found that the assay provides the most reliable and reproducible results when the tissue is maintained in culture for ~24 hours, post excision. i.e., after tissue has had an opportunity to stabilize and equilibrate. Beyond P10, the FM1-43 assay is unreliable for WT hair cells; excised hair cells, particularly outer hair cells, die rapidly and thus do not yield reliable data. Recording of mechanotransduction was not warranted in this case, as there was no compelling reason to anticipate a change in the biophysical properties of hair cell transduction in *Syne4*^{-/-} mice. During the first postnatal week, the FM data provide a rapid assay that can allow a general determination of cell viability for large numbers of cells simultaneously. In the case of the *Syne4*^{-/-} mice we found the mutation does not disrupt FM uptake and cell viability. Other genetic mutations, particularly those that affect the hair bundle, do cause deficits in FM uptake.

2- It wasn't clear either why the authors used the PSC route for gene delivery to target auditory hair cells? Did the authors have data with injections through round window membrane? Any information on AAV delivery at later stages would provide key information on therapeutic window.

Response: Inner ear delivery is technically challenging. Therefore, after trying different approaches we decided to pursue with the PSCC delivery route as it was the one that worked best in our hands. This approach has been described before as safe and efficient (Isgrig K & Chien WW, 2018) and we show it to be so in our hands as well (Fig 3, EV Fig 3).

The comment regarding later stages is important and has been addressed in response to a similar comment made by another reviewer (first comment by referee #2).

3- While the experiment in Fig 7 confirms perception followed by action after sound processing, it's not clear if *Syne4* has a direct critical role in the central auditory pathways? To make *Syne4* function in central pathways a critical finding of the paper, the causal versus correlation between loss of *Syne4* and effect on central processing need be clarified. Experiments in young *Syne4*^{-/-} mice using sound stimuli at frequencies and high intensities to overcome the moderate hearing loss at early stages can help bypass the peripheral sensitivity in absence of *Syne4*.

Response: We thank the reviewer for this important comment. In fact, the authors believe the results demonstrate that *Syne4* has no role in the central auditory system. We have revised the text and elaborated on this issue in response to a similar comment made by another reviewer (comment number 4 by referee #1).

4- Page 9: lines 1-2: In sentence "Another mouse showed no recovery and was excluded from the DPOAE analysis." Why this mouse showed no recovery and why was it excluded ?

Response: Since the injection into the inner ear is technically challenging, occasional failures are expected. We believe that this is the case for two outliers we observed in our data. In one of them we stained the organ of Corti with an anti-FLAG antibody and observed no staining, while in other mice in the treatment group, the same antibody showed clear staining of the nuclear periphery (Fig 3). We have revised the text to state this more clearly:

For two mice there was no evidence of success of the injection, as evaluated by immunofluorescence analysis of Syne4 expression or ABR/DPOAE recovery, and thus they were excluded from downstream analyses

5- Page 3: in sentence " ... proteins such as kinesins and dynemin..." : "dynein" instead of "dynemin"

Response: Corrected

6- Fig.1 in unnecessary as main Figure. If needed, Fig1B can be added to current Figure 2.

Response: Given the broad readership of EMBO Molecular Medicine, we believe it is important to provide some anatomical orientation of the cells being inspected, the delivery route, etc. However, we are happy to defer to the editor's preference on this issue.

7- Fig. 3F: to illustrate further the localization of Syne4 after AAV.Syne4 delivery, I encourage the authors to provide a high magnification image showing protein immunolocalization on sagittal sections of OHCs (e.g. similar to OHCs in Fig 4B). Is Flag.Syne4 also expressed in nuclear membranes of vestibular hair cells ?

Response: We thank the reviewer for this comment that has encouraged us to revisit the data and provide a much clearer figure. Figure 3 now includes a higher magnification inset (3F), a quantification of FLAG fluorescence showing its accumulation at the nuclear periphery (3G) and a 3D surface projection showing the distribution of FLAG staining in the OHC (3H). Figure 4B has also been revised in response to a comment by another reviewer (comment number 1 by referee #1). As these mice show no vestibular phenotype (Horn et al, 2013; Fig EV2 C) and no vestibular phenotype was observed in response to our treatment (Fig EV3 E), we did not see study the vestibular organs of these mice any further.

Figure legend:

F Staining for FLAG at P14 of the organ of Corti of a mouse injected at P1 with AAV.Syne4.

G Quantification of FLAG and DAPI fluorescence intensity along a line centered at the nuclear envelope. Eight OHC were measured.

H 3D surface projection of two adjacent OHC from a mouse injected with AAV.Syne4. In the left cell myosin VIIa and DAPI signals were removed to only show FLAG staining.

Methods:

For FLAG and DAPI intensity a line was centered at nuclear periphery, defined by maximal DAPI fluorescence. Intensity was normalized to the maximal signal detected in each channel.

8- For sake of clarity, "AAV9" instead of "AAV" should be used throughout, including in figures. Also, replace "Mut" with "Syne4^{-/-}"; there is a mixed use of WT or wild-type in legends: homogenize throughout the text;

Response: We thank the reviewer for this comment. When describing the vectors we used we specify their exact components and what we mean by “AAV.Syne4” and “AAV.GFP”, and an illustration of their structure also appears in Fig 3A:

We cloned turboGFP into an AAV2 backbone, downstream of a CMV enhancer and promoter and upstream to a bGH poly-A sequence, and packaged the construct into AAV9-PHP.B capsids (termed AAV.GFP). We then cloned the coding sequence (CDS) of Syne4 into an AAV2 backbone, downstream of a CMV enhancer and promoter, added a 3XFLAG epitope sequence at the 5' end of the Syne4 CDS and a bGH poly-A sequence at the 3' and packaged the construct into AAV9-PHP.B capsids (termed AAV.Syne4).

Also in the Methods:

AAV2.CMV.3xFLAG.Syne4.bGH and AAV2.CMV.turboGFP.bGH vector plasmids were cloned and transferred together with AAV9-PHP.B plasmid to the Vector Core for production of AAV2/9.PHP.B.CMV.3xFLAG.Syne4.bGH and AAV2/9.PHP.turboGFP.bGH

We chose to refrain from using “AAV9-PHP.B.Syne4” or “AAV9-PHP.B.turboGFP” throughout the manuscript for ease of read. Using only “AAV9” would be misleading as readers might think we used the natural AAV9 serotype.

We have revised the text and figures to only include “WT” and “Syne4^{-/-}” instead of “Mut”.

Additional note:

We added an additional grant that has provided funding for the work.

6th Nov 2020

Dear Karen,

Thank you for the submission of your revised manuscript to EMBO Molecular Medicine. We have now received the enclosed report from the three referees who were asked to re-assess it. As you will see the referees are now overall supportive and I am pleased to inform you that we will be able to accept your manuscript pending the following amendments:

1. Please address Referee #2' concerns regarding Fig 3B and C.

2. In the main manuscript file, please do the following:

- remove the red color font
- figure callouts: Figure 1 A & B are not called out, please fix.
- Please update callout from EV Movie 1 to Movie EV1.
- Please remove Movie EV legend from the MS.
- in the Reference: Please remove 'Available at'.
- in Material and Methods, for animal work, gender and age must be indicated.
- in Material and Methods, the statistical paragraph should reflect all information that you have filled in the Authors checklist, especially regarding randomisation, blinding, replication.
- indicate in legends exact n= and exact p= values, not a range, along with the statistical test used. Some people found that to keep the figures clear, providing an Appendix table Sx with all exact p-values was preferable. You are welcome to do this if you want to.

3. Lower right panel of Figure EV1 (32Hz) contains a "splice", can you clarify what it could be and provide source data for this image?

4. Authors checklist: Both co-corresponding authors' names should be on the checklist.

5. Please add a "Data availability" section (before "Acknowledgements" and after "Material and Methods") and include the following single sentence in this section- "This study includes no data deposited in external repositories".

6. The Paper Explained: EMBO Molecular Medicine articles are accompanied by a summary of the articles to emphasize the major findings in the paper and their medical implications for the non-specialist reader. Please provide a draft summary of your article highlighting

7. We would also encourage you to include the source data for figure panels that show essential data. Numerical data should be provided as individual .xls or .csv files (including a tab describing the data). For blots or microscopy, uncropped images should be submitted (using a zip archive if multiple images need to be supplied for one panel). Additional information on source data and instruction on how to label the files are available at

<https://www.embopress.org/page/journal/17574684/authorguide#sourcedata>

8. For more information: There is space at the end of each article to list relevant web links for further consultation by our readers. Could you identify some relevant ones and provide such information as well? Some examples are patient associations, relevant databases, OMIM/proteins/genes links, author's websites, etc...

9. As part of the EMBO Publications transparent editorial process initiative (see our Editorial at <http://embomolmed.embopress.org/content/2/9/329>), EMBO Molecular Medicine will publish online a Review Process File (RPF) to accompany accepted manuscripts.

In the event of acceptance, this file will be published in conjunction with your paper and will include the anonymous referee reports, your point-by-point response and all pertinent correspondence relating to the manuscript. If you do NOT want this file to be published, please inform us.

Please submit your revised manuscript within two weeks. I look forward to seeing a revised form of your manuscript as soon as possible.

Best wishes,
Jingyi

Jingyi Hou
Editor
EMBO Molecular Medicine

*** Instructions to submit your revised manuscript ***

To submit your manuscript, please follow this link:

Link Not Available

1) a .docx formatted version of the manuscript text (including Figure legends and tables)

2) Separate figure files*

3) supplemental information as Expanded View and/or Appendix. Please carefully check the authors guidelines for formatting Expanded view and Appendix figures and tables at <https://www.embopress.org/page/journal/17574684/authorguide#expandedview>

4) a letter INCLUDING the reviewer's reports and your detailed responses to their comments (as Word file).

5) The paper explained: EMBO Molecular Medicine articles are accompanied by a summary of the articles to emphasize the major findings in the paper and their medical implications for the non-specialist reader. Please provide a draft summary of your article highlighting

6) For more information: There is space at the end of each article to list relevant web links for further consultation by our readers. Could you identify some relevant ones and provide such information as well? Some examples are patient associations, relevant databases, OMIM/proteins/genes links, author's websites, etc...

7) Author contributions: the contribution of every author must be detailed in a separate section.

8) EMBO Molecular Medicine now requires a complete author checklist (<https://www.embopress.org/page/journal/17574684/authorguide>) to be submitted with all revised manuscripts. Please use the checklist as guideline for the sort of information we need WITHIN the manuscript. The checklist should only be filled with page numbers where the information can be found. This is particularly important for animal reporting, antibody dilutions (missing) and exact values and n that should be indicated instead of a range.

9) Every published paper now includes a 'Synopsis' to further enhance discoverability. Synopses are displayed on the journal webpage and are freely accessible to all readers. They include a short stand first (maximum of 300 characters, including space) as well as 2-5 one sentence bullet points that summarise the paper. Please write the bullet points to summarise the key NEW findings. They should be designed to be complementary to the abstract - i.e. not repeat the same text. We encourage inclusion of key acronyms and quantitative information (maximum of 30 words / bullet point). Please use the passive voice. Please attach these in a separate file or send them by email, we will incorporate them accordingly.

You are also welcome to suggest a striking image or visual abstract to illustrate your article. If you do please provide a jpeg file 550 px-wide x 400-px high.

10) A Conflict of Interest statement should be provided in the main text

11) Please note that we now mandate that all corresponding authors list an ORCID digital identifier. This takes <90 seconds to complete. We encourage all authors to supply an ORCID identifier, which

will be linked to their name for unambiguous name identification.

Currently, our records indicate that the ORCID for your account is 0000-0002-4913-251X.

Link Not Available

12) The system will prompt you to fill in your funding and payment information. This will allow Wiley to send you a quote for the article processing charge (APC) in case of acceptance. This quote takes into account any reduction or fee waivers that you may be eligible for. Authors do not need to pay any fees before their manuscript is accepted and transferred to our publisher.

Photos 400-800 DPI

*Additional important information regarding figures and illustrations can be found at <https://bit.ly/EMBOPressFigurePreparationGuideline>

The system will prompt you to fill in your funding and payment information. This will allow Wiley to send you a quote for the article processing charge (APC) in case of acceptance. This quote takes into account any reduction or fee waivers that you may be eligible for. Authors do not need to pay any fees before their manuscript is accepted and transferred to our publisher.

***** Reviewer's comments *****

Referee #1 (Comments on Novelty/Model System for Author):

I think this is a nice study that adds to our current knowledge on inner ear gene therapy.

Referee #1 (Remarks for Author):

The revised manuscript has adequately addressed my concerns. I think it is suitable for publication.

Referee #2 (Remarks for Author):

This paper is much improved.

My only concern remains Fig. 3 B and C. The legend claims for B "complete" transduction but the image shows partial incomplete transduction, especially in the IHCs. in C, even in the revised figure,

most IHCs do not express GFP. Not sure how to explain the rest of the data w/o seeing that IHCs are positive. A better documentation is needed, or an explanation of why the rescue occurs even w/o transduction. I am just trying to help the authors to increase the rigor of the observations.

Referee #3 (Remarks for Author):

The revised manuscript is much improved and the authors have responded quite appropriately to most raised issues.

Point by point response

EMM-2020-13259-V2

1. Please address Referee #2' concerns regarding Fig 3B and C.

My only concern remains Fig. 3 B and C. The legend claims for B "complete" transduction but the image shows partial incomplete transduction, especially in the IHCs. in C, even in the revised figure, most IHCs do not express GFP. Not sure how to explain the rest of the data w/o seeing that IHCs are positive. A better documentation is needed, or an explanation of why the rescue occurs even w/o transduction. I am just trying to help the authors to increase the rigor of the observations.

Response: We thank the reviewer for the persistence on this important point and the helpful suggestions. Fig. 3B was imaged at a focal plane showing just the very top of the IHC, where GFP is not seen prominently, in order to show OHC and not supporting cells. In the high magnifications in C we show a lower plane for IHC and a YZ projection. In order to avoid subjectiveness and bias in the interpretation of these results we define cells as positive only when the mean nuclear GFP fluorescence is higher than 2 standard deviations above the mean of the control mouse. It is true that GFP fluorescence in IHC is much lower than OHC, as seen in Fig 3D, but nevertheless these are considered positive by the metric we defined. Since OHC, and not IHC, degenerate in these mice, and since ABR thresholds were correlated with OHC survival and not IHC survival, we believe that rescue is achieved even though GFP levels in IHC are low. We have revised the text to further clarify this point.

In the Results:

“Despite the transduction rate being 100% in all three regions of the cochlea in all three mice examined (Fig 3E), IHC fluorescence was lower than OHC (Fig 3D).”

In the Discussion:

“However, GFP levels in IHC were low in comparison to OHC. It is possible that the rescue we observed should only be attributed to OHC transduction, which would strengthen our hypothesis that Syne4 deafness stems primarily from OHC dysfunction and degeneration.”

In the Methods:

“Mean GFP intensity was calculated in the area of the detected nuclei. An uninjected littermate was used as a control, and the mean fluorescence intensity measured in the nuclei of the control mouse was used to subtract the background. Plots show GFP intensity as arbitrary units normalized to the control.”

2. In the main manuscript file, please do the following:

- remove the red color font

Response: Done

- figure callouts: Figure 1 A & B are not called out, please fix.

Response: Added - A schematic illustration of the ear (Fig 1A), with a focus on the organ of Corti, as well as the timeline of the experiments performed in the study (Fig 1B), are shown.

- Please update callout from EV Movie 1 to Movie EV1.

Response: Done

- Please remove Movie EV legend from the MS.

Response: Done. Added the describing sentence to the relevant section in the Methods.

- in the Reference: Please remove 'Available at'.

Response: Done

- in Material and Methods, for animal work, gender and age must be indicated.

Response: Both males and females were used. Mouse ages were: P8, P10, P12, P14, 4w, 8w, and 12w. For every experiment, the age is specified in the figure legend. We also added this information in some of the figure legends for clarity.

- in Material and Methods, the statistical paragraph should reflect all information that you have filled in the Authors checklist, especially regarding randomisation, blinding, replication.

- indicate in legends exact n= and exact p= values, not a range, along with the statistical test used. Some people found that to keep the figures clear, providing an Appendix table Sx with all exact p-values was preferable. You are welcome to do this if you want to.

Response: Added the following to the Methods: "Littermates were randomized to receive treatment or control. No blinding was performed and all tests were carried out by the same tester. Objective measures were preferred when possible; these include hair cell counts, GFP and FLAG intensity, ABR P1-N1 amplitude and latency, DPOAE thresholds, and all behavioral outputs."

"Two animals were excluded from downstream analysis because they showed no improvement of auditory function. In one of them immunofluorescence confirmed no transgene expression. We interpret this is technical injection failure. All P values are listed in Appendix Table S1."

We also added Appendix Table S1.

3. Lower right panel of Figure EV1 (32Hz) contains a "splice", can you clarify what it could be and provide source data for this image?

Response: This lower right panel is not spliced or cut. A scale bar was added in the image. This image is included in the source data.

4. Authors checklist: Both co-corresponding authors' names should be on the checklist.

Response: Done

5. Please add a "Data availability" section (before "Acknowledgements" and after "Material and Methods") and include the following single sentence in this section- "This study includes no data deposited in external repositories".

Response: Done

6. The Paper Explained: EMBO Molecular Medicine articles are accompanied by a summary of the articles to emphasize the major findings in the paper and their medical implications for the non-specialist reader. Please provide a draft summary of your article highlighting

Response:

Problem

Gene therapy is a promising strategy to treat genetic deafness. Since the auditory systems of humans and mice are very similar in structure, function, and even gene expression, mice serve as an excellent model for basic and translational auditory research. Genetic variants in *SYNE4*, encoding the Nesprin4 protein, have been shown to cause deafness in humans, and *Syne4*-deficient mice show a similar

phenotype. In *Syne4*-knockout mice, the nuclei of outer hair cells (OHC) lose their basal position and degenerate.

Results

We used a gene-replacement approach to rescue hearing in a mouse model of *SYNE4* deafness. This strategy required delivery of the coding sequence of *Syne4* into the inner ears of neonatal *Syne4*-knockout mice by a synthetic adeno-associated virus, AAV9-PHP.B. The results reveal near complete rescue of hair cell morphology and survival, with normalization of auditory function and behavioral responses.

Impact

There are currently over 120 genes associated with inherited deafness. It is of paramount importance to test the feasibility of gene therapy in animal models in order to facilitate the development of future treatments. Our results provide proof-of-concept for the development of gene therapy for *SYNE4* and other forms of deafness.

7. We would also encourage you to include the source data for figure panels that show essential data. Numerical data should be provided as individual .xls or .csv files (including a tab describing the data). For blots or microscopy, uncropped images should be submitted (using a zip archive if multiple images need to be supplied for one panel). Additional information on source data and instruction on how to label the files are available at <https://www.embopress.org/page/journal/17574684/authorguide#sourcedata>

Response: Done.

8. For more information: There is space at the end of each article to list relevant web links for further consultation by our readers. Could you identify some relevant ones and provide such information as well? Some examples are patient associations, relevant databases, OMIM/proteins/genes links, author's websites, etc...

Response: Added after Conflict of interest section

18th Nov 2020

Dear Prof. Avraham,

Please find enclosed the final reports on your manuscript. We are pleased to inform you that your manuscript is accepted for publication and is now being sent to our publisher to be included in the next available issue of EMBO Molecular Medicine.

We would like to remind you that as part of the EMBO Publications transparent editorial process initiative, EMBO Molecular Medicine will publish a Review Process File online to accompany accepted manuscripts. If you do NOT want the file to be published or would like to exclude figures, please immediately inform the editorial office via e-mail.

Please read below for additional IMPORTANT information regarding your article, its publication and the production process.

Congratulations on your interesting work,

Jingyi Hou

Jingyi Hou
Editor
EMBO Molecular Medicine

Follow us on Twitter @EmboMolMed
Sign up for eTOCs at embopress.org/alertsfeeds

***** Reviewer's comments *****

*** ** IMPORTANT INFORMATION *** **

SPEED OF PUBLICATION

The journal aims for rapid publication of papers, using the advance online publication "Early View" to expedite the process: A properly copy-edited and formatted version will be published as "Early View" after the proofs have been corrected. Please help the Editors and publisher avoid delays by providing e-mail address(es), telephone and fax numbers at which author(s) can be contacted.

Should you be planning a Press Release on your article, please get in contact with embomolmed@wiley.com as early as possible, in order to coordinate publication and release dates.

LICENSE AND PAYMENT:

All articles published in EMBO Molecular Medicine are fully open access: immediately and freely available to read, download and share.

EMBO Molecular Medicine charges an article processing charge (APC) to cover the publication costs. You, as the corresponding author for this manuscript, should have already received a quote with the article processing fee separately. Please let us know in case this quote has not been received.

Once your article is at Wiley for editorial production you will receive an email from Wiley's Author Services system, which will ask you to log in and will present you with the publication license form for completion. Within the same system the publication fee can be paid by credit card, an invoice, pro forma invoice or purchase order can be requested.

Payment of the publication charge and the signed Open Access Agreement form must be received before the article can be published online.

PROOFS

You will receive the proofs by e-mail approximately 2 weeks after all relevant files have been sent to our Production Office. Please return them within 48 hours and if there should be any problems, please contact the production office at embopressproduction@wiley.com.

Please inform us if there is likely to be any difficulty in reaching you at the above address at that time. Failure to meet our deadlines may result in a delay of publication.

All further communications concerning your paper proofs should quote reference number EMM-2020-13259-V3 and be directed to the production office at embopressproduction@wiley.com.

Thank you,

Jingyi Hou
Editor
EMBO Molecular Medicine

Corresponding Author Name: Karen Avraham, Jeffrey Holt

Manuscript Number: EMM-2020-13259